# Progressive changes in phenotype, transcriptome and proliferation capacity characterise continued maturation and migration of intestinal cDCs in homeostasis

Fabian T. Hager[1], Trong Hieu Nguyen[1], Asmae Laouina[1], Lydia Kopplin [1], Anna Andrusaite[2], Susan A. V. Jennings[3], Britta Simons[1], Andrea Leufgen[1], Thomas Clavel [3], Simon Milling [2], Immo Prinz[4,5], Reinhold Förster [4], Thomas Stiehl [6], Oliver Pabst [1] & Vuk Cerovic [1] ✉

Conventional dendritic cells (cDCs) are important antigen presenting cells which link innate and adaptive immunity by transferring antigenic information from peripheral organs to T cells in lymph nodes (LNs). However, despite their central function in the induction of adaptive immune responses, the kinetics and molecular regulation of the cDC life cycle and migration remain poorly understood. Using a variety of in vivo techniques, we examine the kinetics of cDC turnover in the intestine and address the molecular changes throughout the various stages of the cDC life cycle – from tissue entry and differentiation to CCR7 upregulation and subsequent migration into draining LNs. Our data demonstrate that the life cycle of gut cDCs is highly dynamic, characterised by continuous alterations in transcriptome, protein expression and proliferation rates. These progressive changes culminate in cDC homeostatic activation and migration resulting in a resource-intensive daily turnover of up to a quarter of intestinal cDCs and an almost complete daily replacement of the migratory cDC compartment in the mesenteric LN. This high turnover rate ensures that the mesenteric LN maintains an accurate reflection of the intestinal immunological state, supporting rapid adaptation to emerging immune challenges.

Conventional dendritic cells (cDCs) are the primary population of antigen presenting cells (APCs) responsible for the initiation of adaptive immune responses. In particular, cDCs are indispensable for priming of naïve T cells and determine the qualitative nature of the ensuing immune response[1]. As such, cDCs represent the key link between the innate and adaptive arms of the immune system.

cDCs are present in all tissues and are continuously replenished from pre-cDCs circulating in blood. Pre-cDCs are dedicated haemato-poietic precursors which differentiate into functional cDCs[2]. cDCs comprise two phenotypically and functionally distinct subsets, cDC1 and cDC2, which sample the local environment via the expression of subset-specific endocytic and sensing receptors[1]. After a period of

[1]Institute of Molecular Medicine, RWTH Aachen University, Aachen, Germany. [2]School of Infection and Immunity, University of Glasgow, Glasgow, UK. [3]Functional Microbiome Research Group, Institute of Medical Microbiology, University Hospital of RWTH Aachen, Aachen, Germany. [4]Institute of Immunology, Hannover Medical School, Hannover, Germany. [5]Institute of Systems Immunology, University Medical Center Hamburg-Eppendorf, Hamburg, Germany. [6]Institute for Computational Biomedicine and Disease modelling with focus on phase transitions between phenotypes, RWTH Aachen University, Aachen, Germany. ✉e-mail: vcerovic@ukaachen.de

tissue residency, cDCs undergo a process mostly referred to as maturation or activation and switch to a state of decreased antigen uptake and enhanced antigen presentation capacity, characterised by increased surface expression of MHCII and costimulatory molecules[3,4]. Importantly, they also upregulate the CC chemokine receptor (CCR) 7, which is required for cDCs to egress peripheral tissues, enter draining lymphatics and transport peripheral antigens to the local draining lymph nodes (LNs). The process of cDC migration to LNs is essential for the induction of immunogenic[5] or tolerogenic[6] T cell responses to peripheral antigens.

The signals regulating cDC activation and migration have mostly been studied in inflammatory settings. Both can be induced by pathogen-associated molecular patterns (PAMPs), either by direct activation of pattern recognition receptors (PRRs) on cDCs or indirectly via cytokine production[7,8]. However, in addition to this proinflammatory induced migration, cDCs also continually migrate from peripheral tissues under homoeostatic steady-state conditions without overt stimulation. Accordingly, in steady-state, the afferent lymph[9] and LNs contain a large population of MHCII[hi] migratory cDCs[10,11] while single-cell transcriptomic analyses indicate the presence of a small population of CCR7-expressing pre-migratory cDCs in various tissues[12,13]. Homoeostatic cDC migration is thought to be especially important in the intestine, which is exposed to a plethora of foreign antigens derived from food, the microbiota and potential pathogens which requires a delicate balance of appropriate immunogenic or tolerogenic adaptive immune responses[14]. Several studies have characterised molecular mechanisms that impact the migration of cDCs or cDC subsets, including MyD88 signalling[15], mechanical deformation of the nuclear membrane[16] or the uptake of apoptotic material by cDC1s[3]. However, none of these pathways fully account for the continuous cDC efflux and turnover and the exact pathways controlling homeostatic cDC activation and migration have remained elusive. A major difficulty in the study of homeostatic cDC migration is that experimental observations reflect a dynamic system affected by several interconnected processes including tissue seeding, proliferation and cell death in addition to cDC migration.

In this work, we utilise a suite of quantitative methods to assess cDC turnover kinetics and the associated changes in cDC phenotype and transcriptome in the small intestinal lamina propria (SI LP). We demonstrate that the SI LP cDC population is highly dynamic, with 14.4–22.3% of cDCs replaced by newly incoming precursors per day. This equates to a high rate of tissue egress and migration that results in the almost complete daily turnover of the migratory cDC population in the draining mesenteric lymph node (MLN). Moreover, our data show continuous changes in cDC transcriptome, phenotype and proliferative capacity over their life span in the SI LP. We therefore define these changes in terms of ongoing cDC development, followed by a transcriptional reprogramming upon homeostatic activation, shared by both major cDC subsets and conserved across tissues.

## Results

### Steady state cDC migration is accompanied by a transcriptional shift shared by cDC1 and cDC2

To analyse the migration of intestinal cDCs, we made use of a CCR7-GFP knock-in reporter mouse model (Supplementary Fig. 1A, B). Analysis of CCR7[gfp/+] reporter mice revealed that pre-migratory CCR7-GFP[+] MHCII[hi] cells represented 1.5% ± 0.5 of the steady-state SI LP cDC population. Notably, in CCR7[gfp/gfp] mice, which lack the functional CCR7 receptor necessary for cDC migration, this CCR7-GFP[+] population accumulated to 11.5% ± 2.7 of the total SI LP cDC compartment (Fig. 1A). To understand the molecular changes underlying the process of gut cDC development, maturation and ultimately CCR7-mediated migration, total CD45[+] CD11c[+] MHCII[+] CD64[−] cDCs comprising both

cDC1s (CD103[+]CD11b[−]) and cDC2s (CD103[−]CD11b[+] and CD103[+]CD11b[+]) from the SI LP of CCR7[gfp/+] reporter mice were sorted and their transcriptome analysed by scRNAseq (Supplementary Fig. 1C). The dimensionality reduction analysis identified 12 clusters that formed three main groups of cells corresponding to cDC1s, cDC2s and pre-migratory CCR7[+] cDCs (Fig. 1B–D). As expected, all clusters expressed typical cDC markers such as the MHCII transcripts *H2-Aa* and *H2-Ab1*, as well as *Itgax* (encoding CD11c), *Flt3* and *CD24a*. The latter two were absent from cluster 11 which showed high expression of typical macrophage markers (*Cd14*, *Adgre*, *Apoe*, *Lyz1*, *C1qa*), likely representing a small number of contaminating macrophages (Fig. 1B–D/Supplementary Fig. 1D). cDC1s (clusters 7, 8 and 0) were characterised by expression of subset specific genes such as *Xcr1*, *Irf8*, *Clec9a* and *Tlr3* while cDC2s (clusters 1–6 and 10) expressed *Sirpa*, *Irf4* and *Itgam* (Fig. 1B–D/Supplementary Fig. 1D). Among cDC2s, two related subtypes could be distinguished. Cells in clusters 1, 2, 3 and 5 expressed high levels of *Itgae* and *Itgb7*, (encoding integrins αE and β7, which form the marker CD103) as well as *Trem1* and *Cd101* corresponding to the CD103[+] CD11b[+] SI LP cDC2s. This subset represents the majority of SI LP cDC2s and is thought to mainly comprise cells of the cDC2a lineage[1]. Notably, while expression of *Itgae* could be detected throughout the cDC2 clusters, cells in cluster 4 expressed low levels of *Itgb7* as well as high levels of *Lyz2* and *Il22ra2*, corresponding to CD103[−] CD11b[+] cDCs in the SI LP, which are thought to be enriched for cells of the cDC2b lineage[17].

The third major group of cDCs comprised the pre-migratory cDCs which expressed *Ccr7* as well as the transgenic marker *EGFP* and formed a cluster (cluster 9) distinct from all other cDCs, highlighting a major shift in transcriptome as cells undergo homeostatic activation. Overall, the transcriptome of the pre-migratory cDCs in cluster 9 exhibited a general downregulation of integrins, including key cDC subset markers (*Itgax*, *Itgae*, *Itgb7*, *Itgb2*, and *Itgam* encoding CD11b) and a concomitant increase in the expression of genes encoding enzymes with disintegrin domains (*Adam8*, *Adam19*, *Adam23*), suggesting a disengagement from the extracellular matrix, which may enable tissue egress. Interestingly, previous research had indicated that the loss of cell-cell contacts may be one of the factors contributing to homeostatic activation of cDCs[18]. Moreover, pre-migratory cDCs showed a downregulation of cell cycle molecules (*Mki67*, *Top2a*) as well as upregulation of genes involved in cell mobilisation and cytoskeleton rearrangement (*Fscn1*, *Samsn1*, *Swap70*, *Marcksl1*, *Dnah2*), apoptosis (*Casp3*, *Fas*, *Cflar*, *Birc2*), cell cycle inhibition (*Cdkn2b*), and chemoattraction (*Ccl5*, *Ccl22*) (Fig. 1D/Supplementary Data 1). Interestingly, in mouse SI LP, CCR7[+] cDCs showed a lower level of *H2* transcripts but higher levels of surface MHCII, which may reflect earlier observations of intracellular stores of pre-made MHCII in cDCs, which are transported to the cell surface upon activation[19,20]. Notably, while pre-migratory cDCs in cluster 9 showed a general downregulation of subset specific markers, two sub-clusters (9.1 and 9.2) exhibited transcriptomic characteristics corresponding to either cDC1s (*Xcr1*, *Aldh1a2*, *Il12b*, *Ncoa7*, *Itgb8*) or cDC2s (*Nrp2*, *Stat4*, *Tmem176b*, *Il1b*, *Cd1d1*) (Fig. 1E). These data illustrate that acquisition of a pre-migratory state in the gut tissue itself is accompanied by a major change in the transcriptional profile of both major cDC subsets while still retaining subset-specific differences.

Overall, the transcriptional characteristics of both cDC1s and cDC2s in cluster 9 corresponded to the previously characterised transcriptional profiles of homeostatically activated cDC1s in skin and thymus[21]. Indeed, direct comparison of the transcriptomic signature of SI LP cDCs from CCR7[gfp/+] mice with an independently performed scRNAseq analysis of WT SI LP mouse cDCs (Supplementary Fig. 2A–C) as well as published scRNAseq datasets from CCR7[+] mouse hepatic cDC1s[12] and CCR7[+] cDC1s in human gut[22], showed substantial overlap, highlighting that homeostatically activated cDCs utilise a conserved migration-

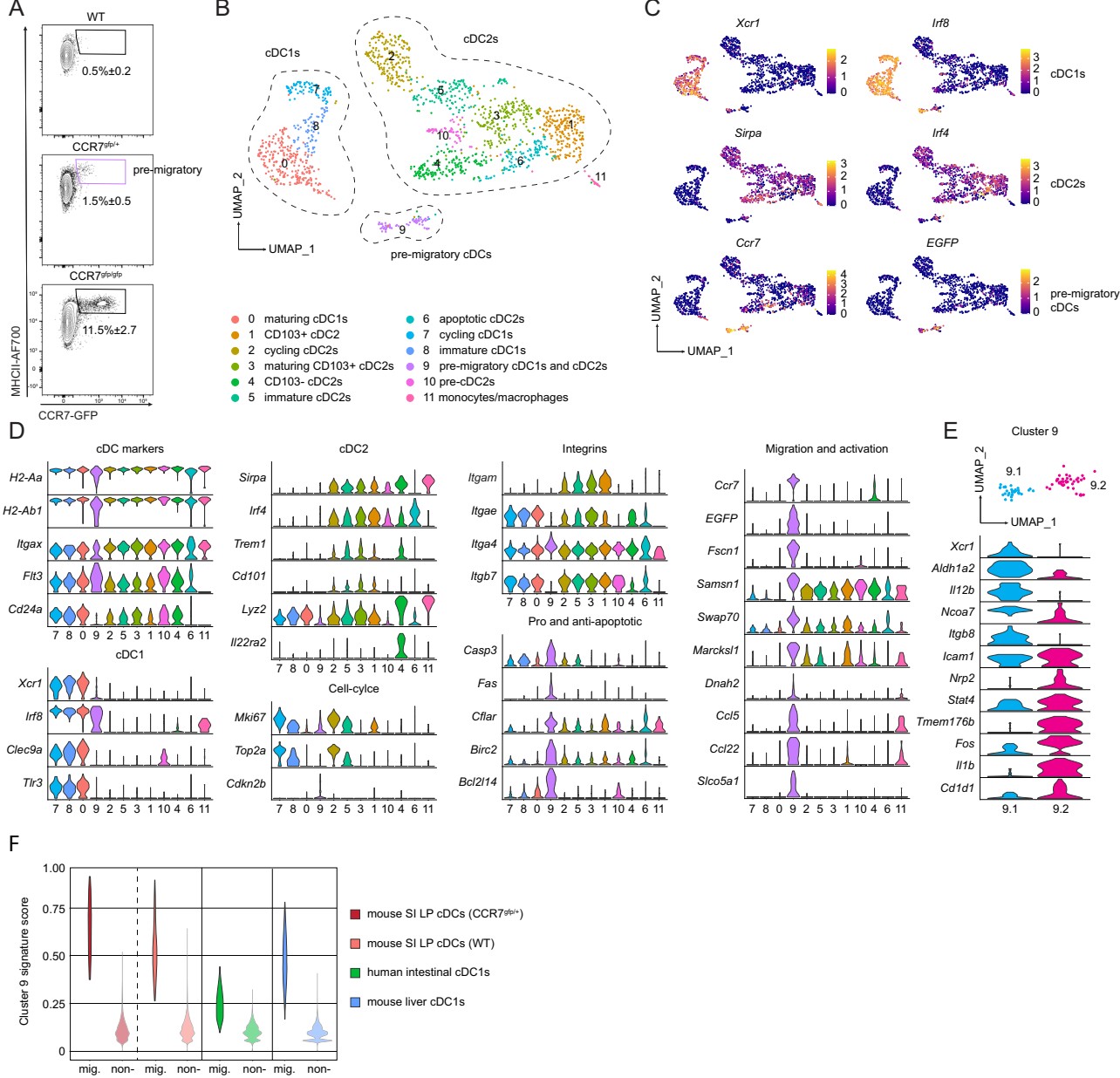

**Fig. 1 | Steady-state cDC migration is accompanied by a major shift in transcriptome that characterises the migration of both cDC subsets. A** Comparison of GFP expression amongst total SI LP cDCs (live, singles, CD45⁺, CD11c⁺MHCII⁺, CD64⁻ (see Supplementary Fig. 1C)) between a WT (CCR7⁺/⁺), CCR7gfp/⁺ and a CCR7gfp/gfp mouse (n = 3 from three experiments). Source data are provided in the Source Data file. **B** UMAP of scRNA-seq data of sorted SI LP cDCs pooled from two CCR7gfp/⁺ mice. **C** Heatmaps represent the expression of selected genes in single cells overlaid on the UMAP analysis from B. **D** Violin plots of selected genes amongst the scRNA-seq clusters from (**B**). **E** Top: Magnified cluster 9 from panel B after sub-clustering. Bottom: Violin plots of selected differentially expressed genes between sub-clusters 9.1 and 9.2. **F** Signature score comparison of migratory cDC cluster 9, with migratory cDCs from an independent mouse intestinal validation cohort (see Supplementary Fig. 2) and migratory cDC1s from publicly available datasets of mouse liver[12] and human intestine[22] (www.gutcellatlas.org) and their non-migratory clusters (Supplementary Data 3).

associated transcriptome across tissues in both mouse and human (Fig. 1F/Supplementary Data 2, 3). Moreover, key features of the pre-migratory cDC transcriptome were also observed in both cDC1 and cDC2 populations of migrating lymph-borne intestinal cDCs (Supplementary Fig. 2D[23]). Interestingly, while CCR7 upregulation correlated with a major shift in transcriptome, the bulk of cDC1 and cDC2 populations comprised a range of sub-clusters, suggesting substantial heterogeneity even prior to activation. We therefore further examined the functional and phenotypic heterogeneity of SI LP cDC subsets prior to CCR7 expression.

## cDC proliferation is induced upon tissue entry and diminished with the initiation of migration

A notable feature of SI LP cDCs were high numbers of cells with the transcriptomic signature consistent with proliferation (expressing cell cycle associated genes such as *Mki67, Top2a, H2afx* and *Tuba1b*) which formed distinct clusters among both cDC1s and cDC2s (clusters 7 and 2, respectively) (Fig. 2A). However, this proliferation signature was absent from both subsets of the pre-migratory CCR7⁺ cDCs (Cluster 9). Moreover, CCR7⁺ cDCs showed a significant increase in expression of cell cycle inhibitor *Cdkn2b*, suggesting a loss of proliferative capacity in pre-migratory cDCs (Fig. 1D). Velocity trajectory analysis of the cDC

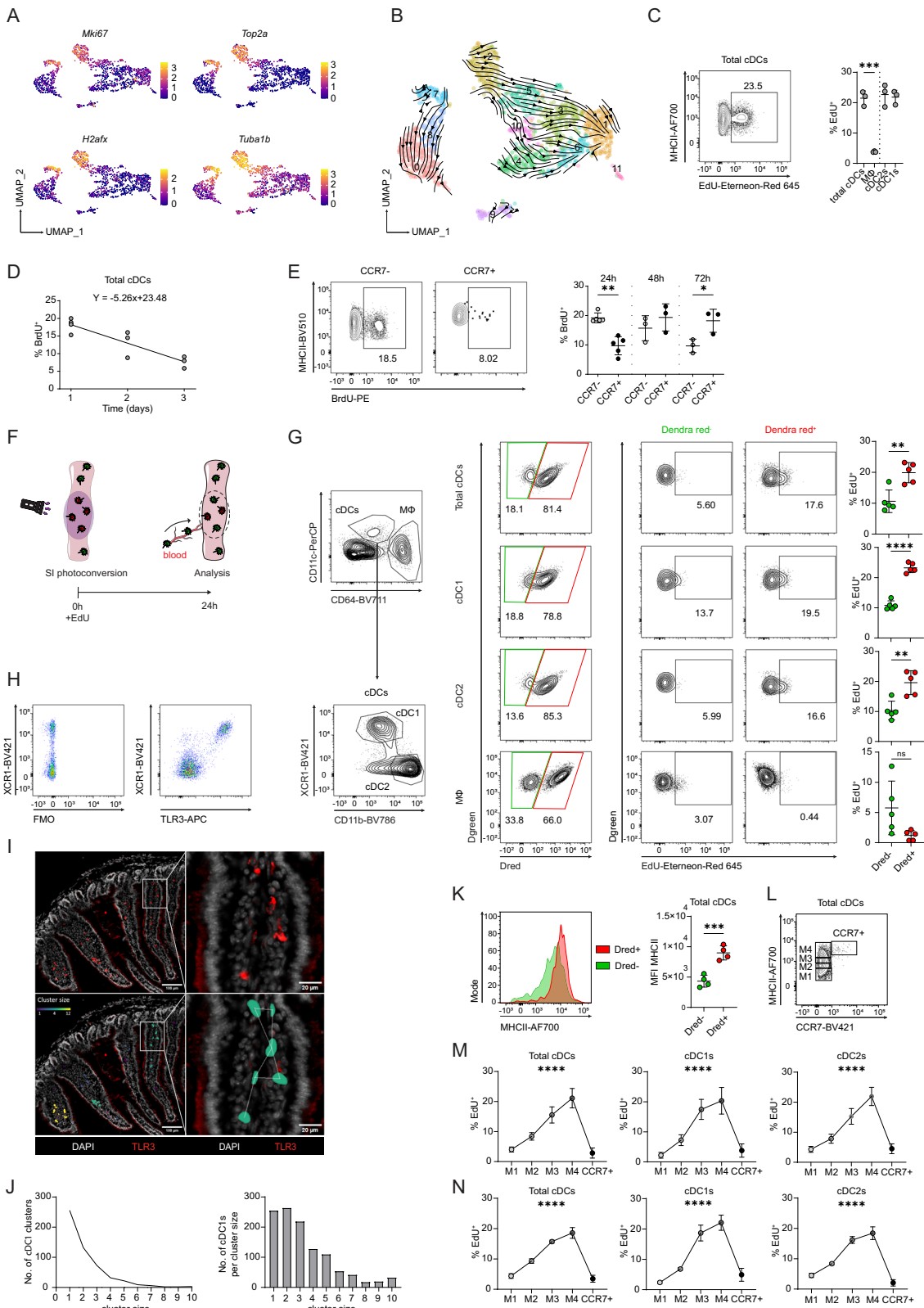

transcriptome suggested that actively cycling clusters among cDC1 and cDC2 represent the earliest developmental stage of cDCs (Fig. 2B).

To confirm that the proliferative capacity of cDCs changes with maturation, we directly assessed the proliferation of SI LP cDC sub-populations by analysis of ethynyldeoxyuridine (EdU) or bromodeoxy-yuridine (BrdU) incorporation 24 h after intraperitoneal (i.p.) injection. The SI LP cDC1s and cDC2s showed similar levels of EdU incorporation,

which was significantly higher compared to intestinal macrophages, the other major APC population in the gut (Fig. 2C). Similar results were observed in a pulse-chase BrdU labelling experiment, with the proportion of BrdU-labelled cDCs decreasing by 5.26% every 24 h after BrdU administration (Fig. 2D). This rate of decrease, equating to a loss of 26.28% of cDCs per day, represents an estimate of the rate at which cDCs egress the intestinal tissue (Supplementary Fig. 3A). Notably, in

**Fig. 2 | cDC proliferation is induced upon tissue entry and decreases upon CCR7 upregulation. A** Heatmaps representing relative expression of selected proliferation associated genes. **B** Proposed cDC differentiation pathway represented by RNA velocity vectors projected on the UMAP from Fig. 1B using scvelo. **C** Left: Representative flow cytometry plot showing the frequency of EdU⁺ cells amongst total SI LP cDCs, 24 h after i.p. injection of 1 mg EdU. cDCs were pre-gated as live, leucocytes, single cells, CD11c⁺MHCII⁺, CD64⁻. Right: Frequency of EdU⁺ cells amongst total cDCs, cDC subsets and macrophages (MΦ) 24 h after i.p. injection of 1 mg EdU. Data are presented as mean ± SD with each dot representing an individual biological replicate (n = 3). Statistical comparison was performed using a two-tailed Student´s *t*-test. **D, E** CCR7^gfp/+ mice were administered 1 mg BrdU by i.p. injection. BrdU incorporation was analysed 24, 48 or 72 h later amongst total SI LP cDCs (**D**) or amongst CCR7⁻ and CCR7⁺ SI LP cDCs (**E**) Data are presented as mean ± SD with each dot representing an individual biological replicate (n = 5 for 24 h from two experiments, n = 3 for 48 h and 72 h from one experiment) **E** The FACS plots show example staining at 24 h. The chart on the right shows the percentage of BrdU⁺ cells among CCR7⁻ (grey) and CCR7⁺ (black) SI LP cDCs. Statistical comparison was performed using a two-way ANOVA with Šídák's multiple comparisons test. **F** Schematic illustration of the experimental setup. Generated in part using images adapted from Servier Medical Art (https://smart.servier.com/), licensed under CC BY 4.0 (https://creativecommons.org/licenses/by/4.0/). **G** Gating strategy and quantification of EdU incorporation amongst Dendra red⁻ (Dred⁻) and Dendra red⁺ (Dred⁺) cDCs and macrophages (MΦ). Cells were gated as live leucocytes, single cells, CD11c⁺MHCII⁺, and either as CD64⁺ macrophages or CD64⁻ cDCs. cDC subsets were further identified by differential expressions of XCR1 (cDC1s) and CD11b (cDC2s). n = 5, each dot represents an individual biological replicate, calculated as the mean percentage of EdU⁺ cells in the same population from two different segments of the same intestine. Data are presented as mean ± SD. Statistical comparison was performed using a two-tailed Student´s *t*-test. **H** Flow cytometry plots showing SI LP cDCs (live, single cells, CD45⁺, CD11c⁺MHCII⁺, CD64⁻) stained with either XCR1 (left) or with XCR1 and TLR3 (right). **I** Representative immunofluorescence images (left) and magnified areas of interest (right) of proximal SI sections stained with DAPI (grey) and anti-TLR3 (red) before (top) and after (bottom) applying Delaunay clustering with a 40 μm threshold. Colour code represents the number of cDC1s per cluster (cluster size). Scale bar lengths are indicated in the images. **J** Quantification of cDC1 (TLR3⁺) clusters and the number of cDC1s per cluster size (35 SI sections from 3 WT mice). **K** Example histogram and quantification of the mean fluorescent intensity (MFI) of surface MHCII amongst Dred⁺ and Dred⁻ SI LP cDCs 24 h after photoconversion. Data are presented as mean ± SD with each dot representing an individual biological replicate (n = 4). Statistical comparison was performed using a two-tailed Student´s *t*-test. **L** Gating strategy used for binning of the SI LP cDC population according to the surface MHCII expression (~25% of the CCR7⁻ cDC population per bin M1-M4) and CCR7 expression. **M, N** Quantification of the frequency of EdU⁺ cells, 2 h after i.p. injection of EdU, in bins representing cDC maturation (as shown in L) for total cDCs, cDC1s and cDC2s in WT and CCR7^gfp/+ (**M**) or CCR7^gfp/gfp mice (**N**). Data are presented as mean ± SD with each dot representing the mean of 7 (**M**) or 4 (**N**) biological replicates. Statistical comparison was performed using an ordinary one-way ANOVA. In all panels, asterisks indicate statistical significance (ns= not significant; * P ≤ 0.05; ** P ≤ 0.01; *** P ≤ 0.001; **** P ≤ 0.0001). Source data and exact *P*-values are provided in the Source Data file.

line with the transcriptomic data, the proportion of proliferating cells was significantly lower among MHCII^hi CCR7⁺ cDCs compared to the CCR7⁻ compartment 24 h post BrdU administration, but continued to increase over the next 2 days, indicating that CCR7⁻ cDCs continue to undergo homeostatic activation, upregulate CCR7 expression and enter the pre-migratory state (Fig. 2E). Collectively, these data further suggest that initiation of cDC migration in the steady state inversely correlates with their proliferative capacity.

However, in light of the high rates of EdU/BrdU incorporation of SI LP cDCs (Fig. 2C, D, E) it remained uncertain to what extent this reflected genuine local proliferation as opposed to labelling of cDC precursors in blood and/or bone marrow prior to tissue entry, as suggested by the velocity analysis (Fig. 2B). To assess the relative contribution of newly incoming and local cDCs to the overall observed proliferation rate, we utilised transgenic mice expressing the histone-fused form of the photoconvertible Dendra2 protein in leucocytes (Vav-H2B-Dendra2)[24]. Following laparotomy, photoconversion of defined segments of the SI was performed by exposure to violet light, altering the emission wavelength of all Dendra-2 expressing cells from green to red. EdU was administered i.p. directly after surgery to label proliferating cells. This system allowed for precise quantification of the relative proportion of EdU⁺ cDCs present in the gut at the time of photoconversion (Dred positive) and newly incoming cDCs (Dred negative) (Fig. 2F).

Somewhat surprisingly, proportions of EdU⁺ cells among newly incoming (Dred⁻) cDCs were significantly lower than that of (Dred⁺) cDCs already present in the tissues at the time point of photoconversion (Fig. 2G). This was true for both cDC1s and cDC2s which showed comparable proliferation rates (Fig. 2G). In line with these observations, immunohistochemistry of SI LP showed that, although some were solitary, the majority (77%) of cDC1s were in proximity (<40 μm distance) to another cDC1, forming clusters of up to 10 cells in size, consistent with a high degree of local proliferation (Fig. 2H-J)[25]. Moreover, 73.5% of cDC1s in clusters consisting of at least five cells were positive for the proliferation marker Ki67, compared to only 47.6% of cDC1s present in small clusters (1–2 cells) (Supplementary Fig. 3B). Unfortunately, we were not able to verify these observations for cDC2s due to technical limitations of their unequivocal identification by immunohistochemistry.

Collectively, our data unambiguously show that cDCs undergo local proliferation in the SI LP. Moreover, our results demonstrate that cDC proliferation is not confined to incoming precursors or earliest developmental stages, but is in fact increased upon tissue entry, likely in response to local environmental cues. In contrast, the analysis of macrophage populations showed the opposite pattern, as expected[26], with the majority of EdU incorporation attributable to the newly incoming Dred⁻ macrophage precursors and only minimal detection of EdU label among resident Dred⁺ macrophages (Fig. 2G).

Interestingly, the EdU incorporation data indicate that, contrary to the initial developmental trajectory analysis (Fig. 2B), cycling cDCs observed in cluster 2 and 7 do not necessarily represent cDC progenitors or immature cDCs at the earliest stage of the cDC life cycle, but may appear as a distinct cluster merely as a consequence of the discrete transcriptional state associated with proliferation. To more precisely assess how proliferation rates change as cDCs mature in the tissue, we sought a surrogate marker that would allow us to subdivide the CCR7⁻ population to reflect ongoing differentiation. The expression of surface MHCII had previously been suggested to correlate with cDC maturation[3,27]. Indeed, newly incoming Dred⁻ cDCs showed lower surface MHCII expression compared to Dred⁺ cDCs already present in the tissue (Fig. 2K). Furthermore, cDC1s and cDC2s showed increased expression of MHCII transcripts along the suggested differentiation trajectories of CCR7⁻ cDCs (Figs. 1D and 2B). Therefore, we divided the CCR7⁻ SI LP cDC population into four equal bins (each accounting for ~25% of the total) according to their surface MHCII expression in addition to the CCR7⁺ cells, which showed the highest MHCII surface expression among cDCs (Fig. 2L). Accordingly, we analysed the rates of EdU incorporation along these maturation bins 2 h (Fig. 2M) or 24 h (Supplementary Fig. 3C) after EdU administration. In line with the data from the photoconversion experiments, the proportion of EdU⁺ cells gradually increased with maturation and surface MHCII expression. However, the percentage of EdU⁺ cells declined sharply in the pre-migratory MCHII^hi CCR7⁺ cDCs (Fig. 2M). Crucially, this loss of proliferative capacity was independent of CCR7 signalling, as a similar pattern of EdU incorporation was observed in the SI LP cDCs of CCR7^gfp/gfp mice which lack CCR7 expression (Fig. 2N). While we cannot map the maturation bins 1–4 precisely to the transcriptome-defined

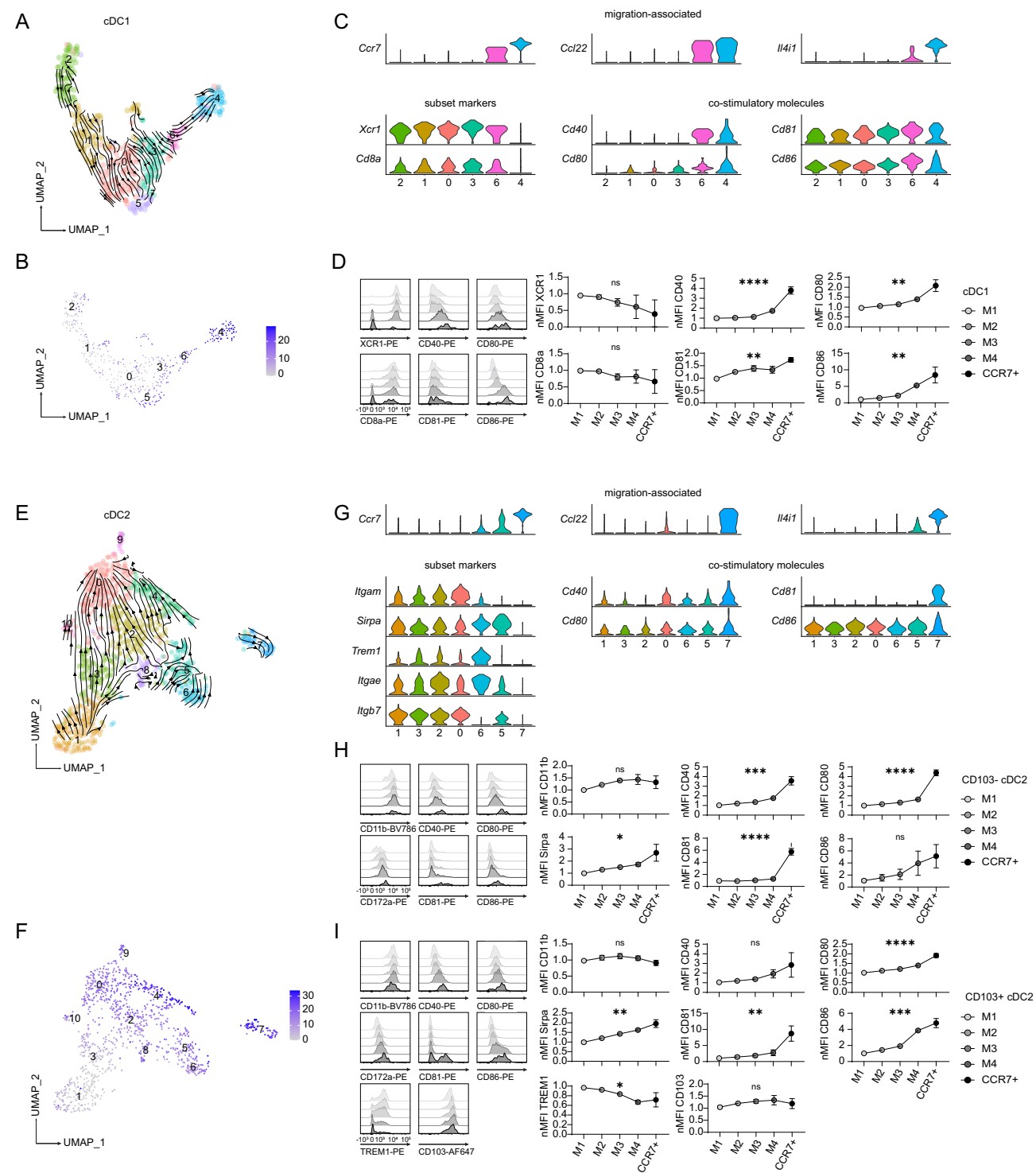

clusters of SI LP cDCs, these data collectively indicate that maturation of cDCs in the tissue is accompanied by an increase in proliferation, while cell cycle inhibition occurs as part of the transcriptional shift to CCR7+ migratory cDCs.

## cDC maturation is accompanied by progressive transcriptional and phenotypic changes preceding homeostatic activation

The gradual change in proliferative capacity of SI LP cDCs suggests that the life cycle of cDCs in tissues reflects ongoing alterations in their biology and functionality, shared between the two major cDC subsets. To understand the maturation-associated transcriptomic and phenotypic changes leading up to the pre-migratory shift in transcriptome in more detail, we analysed and reclustered cDC1 and cDC2 populations separately, and directly compared the changes in their transcriptome prior to CCR7 upregulation.

After reclustering of the cDC1 population, the UMAP, RNA velocity (Fig. 3A) and pseudotime (Fig. 3B) analyses showed a gradual change in transcriptome from the immature cDC1s (cluster 1) towards the pre-migratory cDC1s (cluster 4). Cycling cDC1s formed a distinct cluster (cluster 2), reflecting their characteristic transcriptome. Notably, cells in cluster 6 already exhibited expression of genes associated with the migratory cDC signature, including variable expression of *Ccr7*, *Ccl22*, *Il4i1* and others, suggesting cells of this cluster represented the later stages of maturation (Fig. 3A and C/Supplementary Fig. 4A).

**Fig. 3 | cDC maturation is accompanied by a gradual increase of costimulatory molecules. A** Proposed cDC1 differentiation pathway represented by RNA velocity vectors of the reclustered scRNA-seq data corresponding to clusters 7, 8, 0 and 9 from the UMAP in Fig. 1B. **B** Heatmap of the UMAP as shown in A with overlaid pseudotime values. **C** Violin plots showing the expression of selected migration-associated genes, cDC1 subset markers and co-stimulatory molecules on the cells of cDC1 clusters as defined in Fig. 3A. Numbers along the x-axes indicate the clusters identified in A and B, ordered according to the trajectories suggested by the RNA velocity and pseudotime analyses. **D** Flow cytometry analysis of SI LP cDC1 from CCR7[gfp/+] mice gated along maturation bins as in Fig. 2L. Histograms show example staining of the indicated markers. The graphs show the normalised mean fluorescent intensity (nMFI) of selected cDC1 subset markers and co-stimulatory molecules along cDC1 maturation states. Data are shown as mean ± SD of 7 biological replicates pooled from two independent experiments. **E** Proposed cDC2 differentiation pathway represented by RNA velocity vectors of the reclustered scRNA-seq data corresponding to clusters 2, 5, 3, 1, 10, 4, 6 and 9 in the UMAP in

Fig. 1B. **F** Heatmap of the UMAP as shown in E with overlaid pseudotime values. **G** Violin plots showing the expression of selected migration-associated genes, cDC2 subset markers, and co-stimulatory molecules on the cells of cDC2 clusters as defined in Fig. 3E. Numbers along the x-axes indicate the clusters identified in E and F, ordered according to the trajectories suggested by the RNA velocity and pseudotime analyses. **H, I** Flow cytometry analysis of CD103[−] (**H**) or CD103[+] (**I**) SI LP cDC2s from CCR7[gfp/+] mice gated along maturation bins as in Fig. 2L. Histograms show example staining of the indicated markers. The graphs show the nMFI of selected cDC2 subset markers and co-stimulatory molecules along cDC2 maturation states. Data are shown as mean ± SD of 7 biological replicates pooled from two independent experiments. In panels D, H and I, the nMFI of the indicated marker was normalised relative to the MFI value of the M1 maturation bin which was set to 1. Statistical comparison was performed using an ordinary one-way ANOVA. In all panels, asterisks indicate statistical significance (ns= not significant; *P ≤ 0.05; **P ≤ 0.01; ***P ≤ 0.001; ****P ≤ 0.0001). Source data and exact *P*-values are provided in the Source Data file.

---

Additionally, we observed a gradual change in expression of key genes along the proposed maturation trajectory. The expression of genes encoding subset-specific markers such as *Xcr1* and *Cd8a* showed a gradual decrease in the most mature clusters 3 and 6 with lowest expression in the CCR7[+] pre-migratory cDCs in cluster 4. In contrast, we observed a concomitant and gradual increase in the expression of genes encoding proteins involved in cell migration, T cell co-stimulation and activation, including *Cd40*, *Cd80*, *Cd81* and *Cd86* (Fig. 3C). These transcriptomic changes suggest a maturation continuum, culminating in a larger transcriptomic shift accompanied by the upregulation of CCR7 and other migratory markers upon homeostatic activation. Crucially, similar changes could be observed at the surface protein level, as gradually increasing levels of surface MHCII (see Fig. 2L) correlated with an increase in maturation markers and costimulatory molecules, but a decrease in subset-specific marker expression (Fig. 3D).

Notably, analysis of published datasets of cDC1s from mouse liver, as well as human SI and colon, revealed a similar pattern of co-stimulatory molecules increasing along the predicted differentiation and maturation trajectories with highest expression in pre-migratory CCR7[+] cDCs (Supplementary Fig. 4B, C).

Compared to cDC1s, the structure of the cDC2 population was more complex since cDC2s comprise two subtypes: clusters 1, 3, 2 and 0 corresponding to CD103[+] cDC2s and clusters 6 and 5 corresponding to CD103[−] cDC2s. Clusters 8, 9 and 10 expressed high levels of macrophage markers or apoptosis related transcripts and were not included in further analysis (Supplementary Fig. 4D). Similarly to cDC1s, RNA velocity and pseudotime analyses were consistent with gradual development towards a CCR7[+] pre-migratory phenotype in cluster 7 (Fig. 3E, F). Accordingly, the two cDC2 subpopulations showed expression of genes associated with the migratory signature in the most mature clusters of the two cDC2 subtypes, clusters 0 and 5. Moreover, we observed a gradual increase in the mRNA (Fig. 3G) and protein expression (Fig. 3H, I) of costimulatory molecules and the loss of subset-specific markers such as CD11b, TREM1 and CD103 but not CD172a, possibly reflecting a distinct function of this receptor in cDC2 development[28].

Overall, these data suggest that cDCs in peripheral tissues undergo a progressive maturation programme which is conserved between cDC subsets, tissues and species and which culminates in a major shift in transcriptome at the point of tissue egress, with upregulation of costimulatory molecules and downregulation of most subset-specific marker genes.

## Kinetics of cDC turnover in the small intestine

Our transcriptomic and phenotypic characterisation highlights the dynamic and interconnected nature of cDC maturation, proliferation and migration. Nevertheless, such analyses can only provide a

snapshot of the cDC life cycle in tissues. To analyse the kinetics of the intestinal cDC life cycle, we again utilised transgenic mice expressing the photoconvertible Dendra2 protein to quantify intestinal cDC turnover in vivo. Flow cytometry analysis of the SI at 0, 1, 2 and 3 days after photoconversion allowed for precise quantification of the relative proportion of cDCs present at the time of photoconversion (Dred[+]) and newly incoming cDCs (Dred[−]) (Supplementary Fig. 5A). Within the total cDC compartment, the proportion of Dred[+] cells decreased at a constant rate of ~16.8% per day and was replaced by an equivalent proportion of newly incoming Dred[−] cDCs (Fig. 4A). The calculated half-life of the population ($T_{50}$) was 2.9 days which, extrapolating the linear rate of decrease beyond the observed time points, would indicate a complete turnover of the SI LP cDC population in 5.8 days. We next compared the turnover rates of the two major subsets of intestinal cDCs. cDC1s demonstrated faster kinetics than cDC2s, with 22.3% and 14.4% turnover per day and a $T_{50}$ of 2.2 days and 3.4 days, respectively (Fig. 4B). Interestingly, while the total cDC2 population and CD103[+] cDC2s demonstrated a linear turnover rate, the CD103[−] cDC2 population exhibited an exponential decay (Supplementary Fig. 5B). This likely indicates that CD103[−] cDC2s adopt two distinct fates during their differentiation, resulting in faster turnover. While some bona fide CD103[−] cDC2s develop into an activated phenotype and migrate in intestinal lymph[9], other intestinal CD103[−] cDC2s give rise to CD103[+] cDC2s[29]. As an internal control for the efficiency of photoconversion, we also assessed the kinetics of the resident Tim4[+] macrophage population which has been reported to minimally depend on replenishment by blood-borne precursors[30]. The turnover of this compartment was substantially slower with an estimated $T_{50}$ of 28.8 days (Supplementary Fig. 5C).

To integrate the different experimental measurements of ingress and proliferation into a cohesive model of intestinal cDC turnover, we modelled two distinct possibilities of cDC life cycle regulation (Fig. 4C). One model assumed that cDCs leave randomly after tissue entry, with each cDC having an equal chance to divide and equal chance to migrate (Model 1). The second, deterministic scenario, assumed that all cDCs leave after a set time spent in the tissues with an increasing chance of proliferation over time (Model 2). Using our measured parameters, neither model fully explained the kinetics of cDC turnover observed in vivo (Fig. 4C), raising the possibility that there is heterogeneity in the kinetics of cDC migration, possibly also reflecting the observed heterogeneity in proliferative capacity (Fig. 2M, N). Consistent with this hypothesis, a mixed model (Model 3), assuming that the majority of cDCs leave after a set amount of time, but allowing a proportion to leave earlier, showed the best fit to the observed data (Fig. 4C) across a wide range of potential values for the rate of local proliferation (Supplementary Fig. 5D).

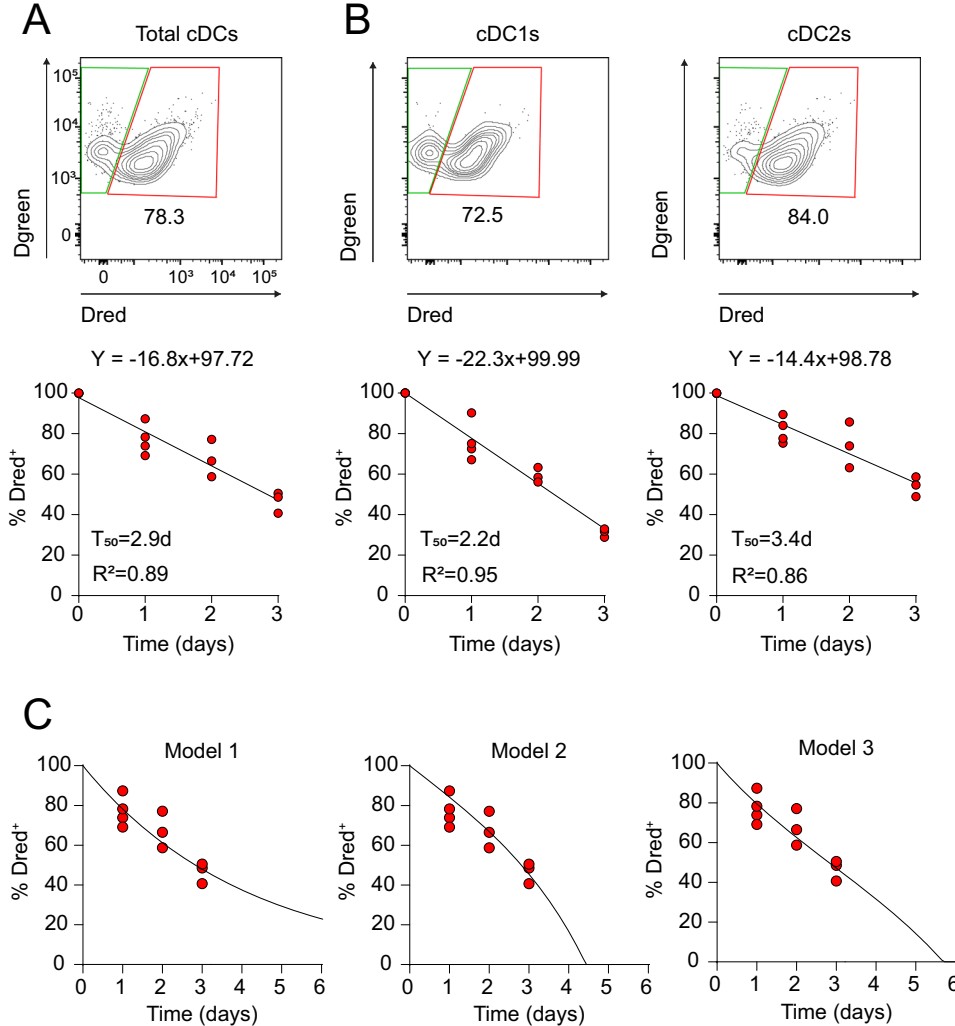

**Fig. 4 | Kinetics of SI LP cDC turnover in the steady-state. A, B** SI segments (~2 cm) of Vav-H2B-Dendra2 mice were photoconverted by exposure to violet light. After 1, 2 and 3 days, total SI LP cDCs (**A**) or cDC subsets (**B**) were analysed for the proportion of Dred⁺ and Dred⁻ cells by flow cytometry (Supplementary Fig. 5A). SI LP cDCs were gated as live leucocytes, single cells, CD11c⁺MHCII⁺ CD64⁻ while the cDCs subsets were identified as CD103⁺CD11b⁻ cDC1s and CD11b⁺ cDC2s. Each dot represents an individual biological replicate (n = 3 for d0, d2 and d3, n = 4 for d1).

The plots show the equation corresponding to the linear regression, and the accompanying $R^2$ value. $T_{50}$ = time at which 50 % of the cells are replenished by Dred⁻ cells. **C** Fit of the different models to the photoconversion data. In model 1 the proliferation rate is assumed to be 0.1/day, corresponding to a population doubling time of ~1 week. In models 2 and 3 the proliferation rate is proportional to the time the cDCs have spent in the SI. The proportionality factor is 0.1/day². Source data are provided in the Source Data file.

## Steady state cDC migration from the intestine results in daily turnover of MLN cDCs

MLN cDCs comprise lymph-borne migratory cDCs, which originated in the intestine and a population of resident blood-borne cDCs, which can be differentiated based on surface expression of CD11c and MHCII[10,11]. Accordingly, CD11c⁺ MHCIIʰⁱ cDCs expressed high levels of CCR7-GFP, which was not expressed by the CD11c⁺ MHCIIⁱⁿᵗ resident cDCs (Fig. 5A). To establish how the observed daily turnover of SI LP cDCs reflects the composition of the migratory MLN cDC compartment, we extended our Dendra2 based in vivo photoconversion approach to directly assess the turnover kinetics of MLN cDC subsets (Fig. 5B, Supplementary Fig. 6A). Flow cytometry analysis of MLNs at 0, 4, 8, 12, 18 and 24 h after photoconversion allowed for precise quantification of the relative proportion of cDCs present at the time of photoconversion (Dred⁺) and newly incoming cDCs (Dred⁻).

The data revealed the remarkably rapid turnover of the small intestine-draining MLNs (sMLN) migratory cDC population, which could be modelled by exponential decay curves (Fig. 5C). Over 80% of the migratory cDC compartment of the sMLN was replaced within 24 h and over 98% replaced by 48 h. The migration kinetics of cDC1s, as well

as CD103⁺ and CD103⁻ cDC2 subsets were similar with the $T_{50}$ of 9.5, 6.3 and 10.1 h respectively. This was in stark contrast to the sMLN resident cDCs, which had a $T_{50}$ of 6.6 days, reminiscent of SI LP cDCs (Fig. 5D). The short lifespan of migratory MLN cDCs is consistent with the upregulation of apoptotic genes in the SI LP pre-migratory cDCs (Fig. 1D, Supplementary Data 1). Accordingly, migratory MLN cDCs showed a significantly higher proportion of cleaved caspase 3⁺ cells compared to the resident cDC compartment (Fig. 5E).

Somewhat surprisingly, despite higher exposure to microbial stimuli, similar migration kinetics of migratory cDCs were also observed in the MLNs draining the large intestine (Supplementary Fig. 6B). In line with this, microbial colonisation status had no effect on the relative proportion of pre-migratory cDCs in the SI LP or migratory cDCs in the MLN (Supplementary Fig. 6C, D).

In summary, our data show that the highly dynamic turnover of SI LP cDCs drives a high rate of homeostatic cDC migration to the MLN resulting in the almost complete daily replacement of the entire lymph-borne cDC population. Therefore, the migratory cDC compartment exists in a constant state of turnover, continually acquiring, transporting and presenting new antigen from the periphery.

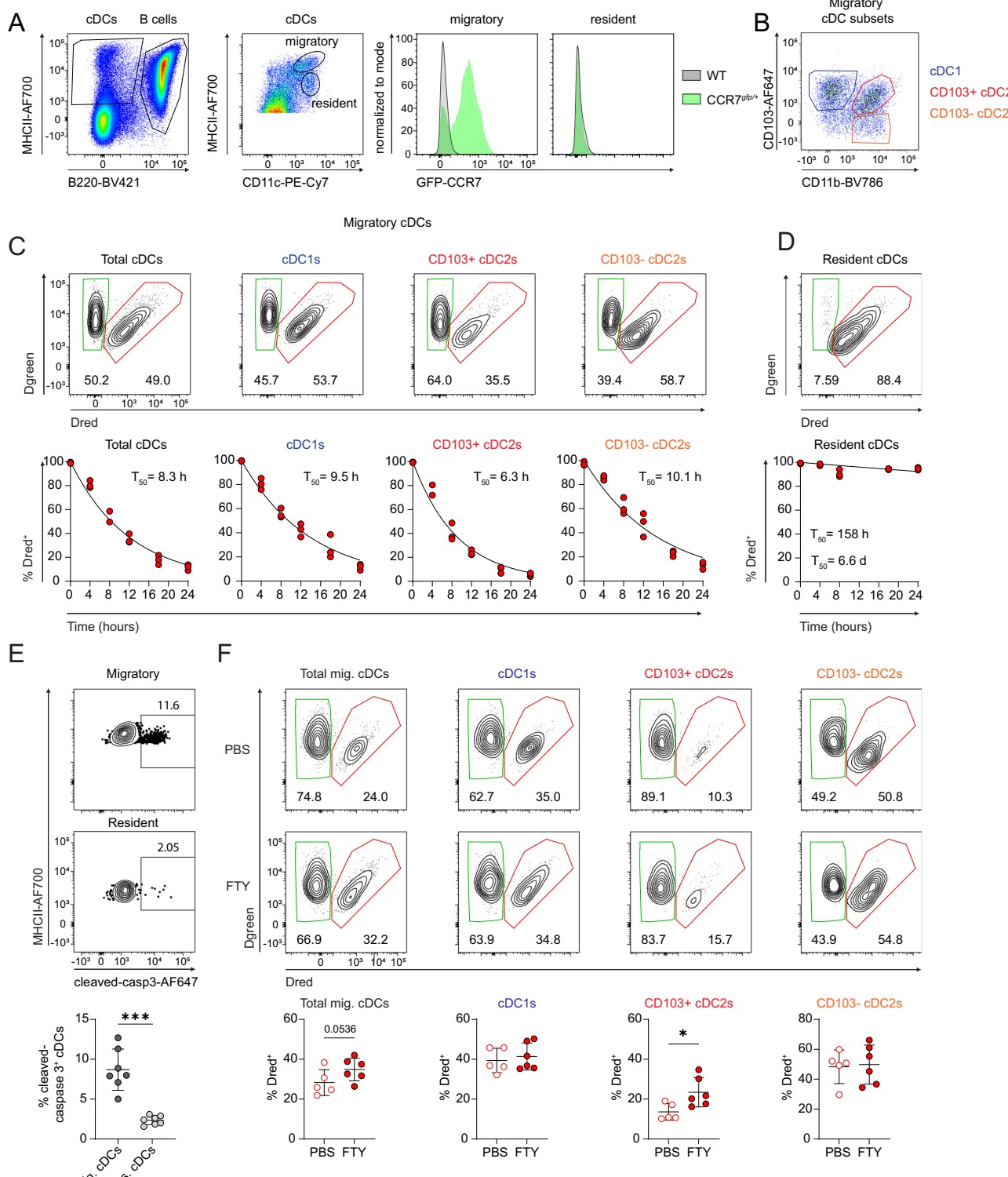

## Inhibition of S1P signalling slows CD103⁺ cDC2 migration kinetics

Previous work by us and others showed that treatment with the sphingosine-1-phosphate (S1P) functional antagonist FTY720 could inhibit steady-state cDC migration[31,32]. Furthermore, both cDC subsets collected from small intestinal draining lymph show an upregulation of S1P receptors (S1PR) suggesting a possible role for this receptor family in cDC migration (Supplementary Fig. 2D). To directly test whether S1PR signalling could affect the kinetics of intestinal cDC migration, Vav-H2B-Dendra2 mice were administered FTY720 for 3 days, followed

by photoconversion of the MLN chain and analysis 16 h later. Effectiveness of the FTY720 treatment was confirmed by a significant increase of Dred⁺ B cells in the sMLN compared to PBS-treated controls (Supplementary Fig. 6E). In accordance with published data[31], FTY720 treatment induced an increase in the proportion of Dred⁺ migratory cDCs, indicating a slower turnover. Interestingly, subset specific analysis demonstrated that this decrease in migration kinetics was entirely due to the effect of FTY720 on the CD103⁺ cDC2 subset in both the SI-draining and colon-draining MLNs, while the kinetics of cDC1 and CD103⁻ cDC2s were unaffected (Fig. 5F/Supplementary Fig. 6F).

**Fig. 5 | Homeostatic migration of SI cDCs results in an almost complete daily turnover of the sMLN migratory cDC compartment. A** Left: Gating strategy for B220⁻ migratory (CD11c⁺MHCII^hi) and resident (CD11c⁺MHCII^int) cDCs in lymph nodes. Right: Exemplary histograms displaying the CCR7 expression of migratory and resident cDCs in the small intestinal draining lymph node of a CCR7^gfp/+ reporter mouse. **B** Representative flow cytometry plot showing the gating for cDC subsets in the migratory cDC compartment of the small intestinal draining lymph node (sMLN). Cells were gated as live leucocytes, single cells, MHCII⁺B220⁻. **C, D** The MLN chain of Vav-H2B-Dendra2 mice was photoconverted by exposure to violet light. After 4, 8, 12, 18 and 24 h, sMLN migratory (**C**) or resident (**D**) cDCs were analysed for the proportion of Dred⁺ and Dred⁻ cells by flow cytometry. Migratory cDC subsets were identified as CD103⁺CD11b⁻ cDC1s and the CD11b⁺ cDC2 compartment was split into CD103⁺ and CD103⁻ cDC2 subpopulations (see panel B). Each dot represents an individual biological replicate with n = 3 for each timepoint.

T₅₀ = time after which 50% of the cells were replenished by Dred⁻ cells. **E** FACS plots show example staining of migratory and resident MLN cDCs (gated as in A) for cleaved caspase 3. The graph shows the percentage of migratory and resident MLN cDCs stained for cleaved caspase 3. Data are presented as mean ± SD. Each dot represents an individual mouse (n = 7) pooled from two independent experiments. Statistical comparison was performed using a paired two-tailed Student´s *t*-test (*** P ≤ 0.001). **F** Representative gating and quantification of Dred⁺ migratory cDCs and cDC subsets in the sMLN after 3 days of consecutive i.p. administration of 1 mg/kg FTY720 (FTY; filled symbols) or PBS (empty symbols) and 16 h after photoconversion of the MLN. Data are shown as mean ± SD. Each dot represents an individual mouse (PBS control group (n = 5); FTY720 group (n = 6)) from two independent experiments. Statistical comparison was performed using a one-tailed Student´s *t*-test (*P ≤ 0.05). Source data and exact *P*-values are provided in the Source Data file.

Notably, analysis of intestinal migratory lymph cDCs detected an increase in *S1pr1, S1pr3, S1pr4* and *S1pr5* in both cDC1s and cDC2 compared to the SI LP (Supplementary Fig. 2D). This suggests that different cDC subsets, while adopting a very similar transcriptomic signature associated with migration, may nevertheless partially utilise distinct mechanisms for some stages of this process and furthermore highlights the usefulness of in vivo cell tracking approaches in quantifying cDC migration and testing candidate pathways involved.

Collectively, combining in vivo techniques and mathematical modelling, we provide quantitative data and in-depth insights into the interplay of intestinal cDC turnover, in situ proliferation and tissue egress. The alacrity of these interlinked processes, even in steady-state, ensures that antigenic information in the MLN continually reflects the current immunological status of the gut and allows for a prompt responsiveness to dynamic immunological challenges.

## Discussion

Migration of cDCs via lymphatic drainage is the key mechanism by which antigen is transported from peripheral tissues to the lymphoid compartments where de novo adaptive responses can be initiated. While cDC migration can be induced by inflammatory stimuli, cDCs also migrate under homeostatic conditions and are thought to be essential in the induction of peripheral tolerance to harmless antigens, particularly in the intestine[14]. However, despite its importance, the exact kinetics and molecular changes leading up to cDC migration are not well understood. In this study, we used a variety of approaches to quantitatively define the kinetics of intestinal cDC turnover in vivo and, in parallel, address the molecular changes that occur at various stages of cDC life cycle - from tissue entry and differentiation to the initiation of the CCR7⁺ pre-migratory phenotype and tissue egress. Our data show that the life cycle of gut cDCs is highly dynamic, characterised by progressive changes in transcriptome, surface protein expression and tissue-induced increase in proliferation rates. These progressive changes culminate in homeostatic activation of intestinal cDCs, accompanied by profound transcriptional reformatting and cell cycle inhibition preceding the egress from intestinal tissue and migration to the draining MLN.

In their pre-migratory state, both subsets of cDCs undergo similar reprogramming—upregulating CCR7 and genes associated with motility, cytoskeletal rearrangement and a general shift in cDC functional specialisation away from a sentinel role in the tissues to antigen presentation in the draining LN, a characteristic transcriptional profile also observed in human intestinal CCR7⁺ cDCs[22]. Collectively, this transcriptional signature of intestinal pre-migratory cDCs is shared across multiple peripheral tissues including liver[12], lung[21], skin[33] and even found in other migrating APC populations[34]. While this signature has also been observed in a population of tumour-associated cells termed mregDCs[35] it is likely that these simply reflect an activated state, rather than a distinct subset identity[36]. Therefore, these transcriptional changes likely represent the fundamental features associated with

homeostatic activation of cDCs/APCs conserved in various tissues and across species[37].

However, while the transition to the CCR7⁺ pre-migratory state showed a common gene expression signature, cDC1s and cDC2s retained differences in transcriptional reprogramming, reflecting their functional specialisation. Pre-migratory cDC1s showed a notable upregulation of Integrin beta-8, which cleaves pro-TGFβ into its active form, and ALDH, the enzymes regulating production of retinoic acid, consistent with the reported role of this subset in the induction of tolerogenic T cells[38,39]. Notably, cDC1s also upregulated *Il12b*, in accordance with the higher potential of cDC1s for IL-12 production and induction of Th1 or CTL responses in the gut[40]. Conversely, pre-migratory cDC2s expressed higher levels of *IL1b*, which may reflect a different functional specialisation[9,27]. Moreover, prior studies had suggested that signals regulating homeostatic migration of cDC1 and cDC2 may differ. For instance, homeostatic activation of cDC1s but not cDC2s can be induced by LXR-mediated signalling following apoptotic material uptake[3]. Previous studies have shown that inhibition of S1P receptors can hinder cDC migration, possibly through the downregulation of surface CCR7[31,32]. Our experiments revealed a differential requirement for S1PR signalling in steady-state migration of cDC2s but not cDC1s. These data suggest the molecular pathways utilised for tissue egress differ between cDC subsets which may reflect the different localisation or developmental requirements of SI LP cDC subsets. Notably, CD103⁺ cDC2s have been reported to establish intimate contact with epithelial cells during their development[41], which might necessitate the usage of additional chemotactic receptors compared to cDC1s residing in the LP.

In addition to the major transcriptional shift associated with homeostatic activation and migration, our data reveal progressive changes in cDC phenotype in tissues. While traditionally maturation refers to the transition from the sentinel and antigen-acquisition state to an antigen-presentation state, we suggest that the term could be expanded to include the progressive changes in the phenotype, transcriptome and proliferative capacity of tissue cDCs, prior to CCR7 upregulation and migration. Indeed, the previously proposed term homeostatic activation may be more appropriate for the sudden shift in functionality associated with CCR7 upregulation and migration[1]. While it has been assumed that the majority of cDC proliferation in tissue is due to newly incoming precursors and the earliest stages of cDC differentiation, our data clearly indicate that cDCs proliferate throughout their maturation in the intestinal tissue, consistent with previously characterised clonal cDC clusters[25]. Furthermore, cDC proliferation capacity increased after tissue entry and throughout their maturation and was only lost immediately prior to tissue egress. This suggests that local microenvironmental factors in the SI LP, such as Flt3L or CSF2, may control cDC development in peripheral tissues[42,43].

In contrast, homeostatic activation of cDCs is accompanied by a loss of proliferative capacity as evidenced by low levels of EdU incorporation. Interestingly, despite the high rate of turnover and

migration, the degree of EdU incorporation of CCR7+ cDCs did not equilibrate with the bulk cDC population even 24 h post EdU administration. This suggests that among the bulk MHCII[hi] CCR7− cDC population, the cDCs least likely to proliferate (and incorporate EdU) are most likely to upregulate CCR7 and subsequently leave. However, whether cDC proliferation and migration are regulated by same stimuli still remains to be determined.

The molecular mechanisms controlling steady-state cDC migration are one of the great open questions in cDC biology. It has long been known that cDC migration is induced by overt inflammatory stimuli such as TLR agonists or proinflammatory cytokines[7,8]. Subsequently, several signals driving cDC migration in the steady state were proposed[15,16,18]. However, disruption of any of these pathways does not fully recapitulate the phenotype of CCR7 deficiency and none of the proposed mechanisms is likely to be the sole determinant of the regular and continuous steady-state cDC migration. Notably, mathematical modelling based on our data suggests that homeostatic cDC turnover may be the result of two different modes of cDC migration. Some cDCs may undergo homeostatic activation and migrate in response to discrete signals, such as those suggested above. Alternatively, cDCs may have a set lifespan in the tissue and subsequently leave, which may be controlled by cumulative signals, e.g. gradual accumulation of phagocytic or efferocytic cargo[3]. Interestingly, a recent report proposed that local availability of growth factors may be an important driver of homeostatic cDC maturation in LNs[4]. Indeed, depletion of local growth-factors may act as a self-regulating stress response driving peripheral cDC maturation while accounting for a low rate of proliferation we observed in pre-migratory tissue cDCs. However, the observed kinetics of cDC turnover and proliferation could not be fully explained by either model, while a mixed model encompassing both behaviours closely fit the observed data. It is therefore likely that cDCs do not uniformly follow a single mode of migration. Instead, while some tissue cDCs stay an average of 5.5–6 days, undergo local proliferation in the SI LP and migrate at the end of this period, others may be induced to migrate earlier, in response to localised activation signals.

The loss of proliferative capacity in pre-migratory cDCs is paralleled by an upregulation of apoptosis- related transcripts. This is also evident in migrating cDCs in lymph and MLN which may account for the extremely fast turnover rates of the migratory cDC compartment in MLNs. Interestingly, CCR7 ligation induces anti-apoptotic signals in DCs, which suggests a mechanism by which cell death of peripheral tissue cDCs may be deferred in order to enhance antigen presentation in the inductive sites[44]. However, it remains elusive whether the upregulation of apoptosis-related genes, may act as a driver of-, or occurs in response to-, homeostatic cDC activation.

Our data provide quantitative information on the key aspect of the intestinal cDC life cycle (Supplementary Fig. 7). We estimate the rate of influx of new cDCs/cDC precursors into the SI LP to be 14.4–22.3% daily. In addition, the number of cDCs in the SI LP is maintained through local proliferation, which increases with cDC maturation. These are balanced by a daily rate of tissue egress which we here estimate to correspond to a minimum of 26.3% of the cDC population. Interestingly, this rate of intestinal cDC turnover is notably higher than previously reported in other tissues[45–47]. The reasons for this are not clear but may be due to higher exposure to bacterial components and PAMPs in the intestine compared to other tissues. However, it should be noted that cDCs in the MLNs draining the large intestine showed similar kinetics to SI-draining MLNs, despite the higher bacterial load and we could detect no decrease in the proportion of CCR7+ migrating cDCs in the SI LP or MLN of germ-free mice. Accordingly, previous studies found that the percentage of migratory cDCs in the MLN or intestinal lymph was not affected in germ-free hosts suggesting that exposure to microbial stimuli is not a critical step in the induction of homeostatic intestinal cDC migration[15,48]. Other possibilities include a general exposure to a high concentration of soluble and particulate antigens as well as antigen derived from apoptotic cells[3]. Alternatively, the high rate of SI LP cDC turnover may also be driven by the relatively limited space available, leading to rapid depletion of local resources[4] and a higher degree of mechanical stress[16] which both may increase the rate of homeostatic activation.

The extremely fast kinetics of SI LP cDC turnover necessitate immense investment of resources and highlight the importance of continued tissue screening and ongoing antigen presentation. These findings also have important practical implications, as the high cDC turnover may make it difficult to manipulate cDC populations in vivo using inducible Cre systems, transfection, cell transfer and others. On the other hand, the fast turnover of cDCs may make them more suitable for manipulation in a therapeutic setting for one-off or even repeated treatments. Their short lifespan could also reduce any potential long-lasting side effects, which may account for the excellent safety record of DC-based therapies.

## Methods

### Mice

Wild-type (WT) C57BL/6 J (CD45.2), Vav-H2B-Dendra2 (Dendra, backcrossed to C57BL/6 J background)[24] and CCR7[gfp/gfp] mice were maintained under specific pathogen free (SPF) conditions at the animal facility of the Uniklinik RWTH Aachen. WT C57BL/6 J mice used for scRNAseq analysis in Supplementary Fig. 2 were maintained under standard animal house conditions at the University of Glasgow. CCR7-GFP reporter mice were generated by the insertion of an IRES- and enhanced green fluorescent protein (eGFP)- containing cassette into the 3' untranslated region of exon 3 of the CCR7 gene, thereby generating a null allele of *Ccr7*. The targeting construct was transfected by electroporation into Bruce-4 C57BL/6 embryonic stem (ES) cells and the successfully targeted ES clones were injected into C57BL/6 blastocysts. Chimaeric mice were bred onto the C57BL/6J background as either heterozygous (CCR7[gfp/+]) or homozygous (CCR7[gfp/gfp]) lines. Overall, the CCR7[gfp/+] mice were indistinguishable from WT mice and had normal lymph node and spleen architecture and cellularity. The CCR7[gfp/gfp] mice had undetectable surface CCR7, smaller lymphoid organs and a lack of migrating cDCs in LNs, consistent with the published phenotype of CCR7 KO mice[49]. All mice were housed at max. groups of 5 in individually ventilated cages at 21 °C and 45–65% humidity with a conventional light cycle of 12 h. All experiments were performed on 8–18-week-old, age- and sex- matched mice, unless otherwise stated. For experiments where different cells within the same mouse were compared, both male and female mice were used. Where two groups of mice were compared, groups were blocked so that equal numbers of male and female mice of similar ages were used in each group. The experiments were approved by the North Rhine-Westphalia State agency for nature, environment and consumer protection (Landesamt für Natur, Umwelt und Verbraucherschutz Nordrhein-Westfalen, LANUV no. 81-02.04.2019.A373). Euthanasia was performed by $CO_2$ inhalation followed by cervical dislocation. All experiments were performed in accordance with the local guidelines and ethical regulations (Tierschutzgesetz).

### Germ-free mice

All experiments on germ-free mice were performed as described previously[50] under Ethical Approval (LANUV no. 81-02.04.2019.A065) in accordance with EU regulation 2010/63/EU. Germ-free mice were bred in germfree (GF) isolators (NKPisotec, Flexible film isolator type 2D) under sterile conditions in gnotobiotic facility A (University Hospital of RWTH Aachen, Germany). Germ-free mice were removed from breeding isolators at 5 weeks of age and housed in HEPA-filtered bioexclusion isocages (Techniplast ISO30P). In both isolators and isocages, Tek-Fresh bedding (ENVIGO) was used. Mice were housed in single sex cages in the same room. Room temperature was kept

between 21–24 °C and 30–70% humidity on a 12 h:12 h day/night cycle. Faecal samples were taken before experiments to confirm the GF status via microscopic observation after Gram-staining and plating on both anaerobic and aerobic agar plates.

### In vivo photoconversion

Prior to photoconversion, Vav-H2B-Dendra2 mice were given analgesic Carprofen (Pfizer, 5 mg/kg of body weight) by i.p. injection and anesthetized by inhalation of Isoflurane (Piramal). Laparotomy was performed by making a 1.5 cm medial skin incision below the sternum followed by an -1 cm incision in the muscle along the *linea alba*. For photoconversion, the BlueWave 75 light curing system (Dymax) with a 390/40 nm band pass filter at an intensity of 120 mW/cm$^2$ was used. For photoconversion of SI segments, the gut was exposed on PBS-soaked cotton swabs and two distinct sections (-2 cm long) were photoconverted 4×15 sec. Any surrounding tissue was covered with aluminium foil to prevent light exposure. For photoconversion of the MLN chain, the mesentery harbouring the MLN was exposed on PBS-soaked cotton swabs and photoconverted 3×15 sec. Post-surgery, the abdominal muscle wall was sutured by continuous stitch using surgical suture (Marlin violet, HR17, catgut) and the skin closed with surgical staples (Fine Science Tools, 9 mm). After the procedure, mice were given Novalgine (Ratiopharm) *ad libitum* (0.8 mg/ml) in drinking water. In some experiments, mice were administered 1 mg/kg of FTY720 (Sigma) i.p. in 100 µl PBS. Efficiency of photoconversion at $t = 0$ was >98%.

### Tissue preparation and cell isolation

Intestinal leucocytes were isolated according to established protocols[51]. Briefly, intestinal tissues were flushed with Hank´s Balanced Salt Solution (HBSS) supplemented with 3% Foetal Calf Serum (FCS) and the Peyer´s patches removed. Intestines were cut into 5 mm sections and washed twice in 2 mM EDTA for 20 min at 37 °C with shaking, then filtered through a 50 µm Nitex mesh (Sefar). The SI tissue was incubated with 1 mg/ml collagenase VIII (Sigma, C2139-1G) in 15 ml RPMI at 37 °C with shaking for 15 min until digestion was complete. For large intestine, tissue segments were incubated with a mix of enzymes (collagenase V, Sigma, C9263-1G, 0.85 mg/ml; Collagenase D, Roche, 11088882001, 1.25 mg/ml; Dispase, Gibco, 17105041, 1 mg/ml; DNase, Roche, 101104159001, 30 mg/ml) at 37 °C with shaking for a minimum of 15 min until complete digestion. Single cell suspensions were filtered through 100 µm cell strainers (Corning), centrifuged at 400 x g at 4 °C for 6 min and resuspended in PBS containing 3% FCS (PBS-FCS) for further analysis

MLNs were excised, cleared of perinodal fat, cut into pieces and incubated in 1 mg/ml Collagenase D (Roche, 11088882001) in RPMI at 37 °C with shaking for 45 min. Cells were then filtered through 50 µm Nitex mesh, centrifuged at 400 x g at 4 °C for 6 min and resuspended in PBS-FCS for further analysis.

### Flow cytometry

Prior to surface staining, single cell suspensions were incubated in PBS-FCS containing 5% rat serum for 10 min at 4 °C. After washing in PBS-FCS, cells were resuspended in the antibody mixture for 30 min on ice. Fluorescent-dye-conjugated combinations of the following anti-mouse antibodies were used: BioLegend: B220 (RA3-6B2, 103203, 103251), CD103 (2E7, 121410), CD11b (M1/70, 101243), CD11c (N418, 117318), CD19 (6D5, 115520), CD3 (17A2, 100236), CD4 (RM4-5, 100555), CD40 (3/23 124610), CD45 (30-F11 103105, 103114), CD45.2 (104, 109806, 109808), CD62L (MEL-14, 104445), CD64 (X54-5/7.1, 139318, 139314), CD80 (16-517 10A1, 104708), CD81 (Eat-2, 104905), CD86 (GL-1, 105008), Ly6C (HK1.4, 128037), MHCII (M5/114.15.2 107635, 107622), Nk1.1 (PK136, 108725), Tim4 (RMT4-54, 130009), XCR1 (ZET 148216, 148203),eBioscience: CD101 (Moushi101, 12-1011-80), BD: CD172a (P84, 560107, 144014), R&D: Trem1 (174031, FAB1187P). Optimal staining

concentration was determined by titration for each vial of antibody. When 7-Aminoactinomycin D (7AAD) (BioLegend, 420404) was used as live-dead dye it was added for the last 5 min into the antibody staining mixture (final concentration of 2.5 µg/ml). When Zombie NIR (BioLegend, 423105) was used as live-dead dye, cells were stained with the PBS+Zombie NIR mix (1:1000) for 10 min at 4 °C in the dark prior to staining with surface antibodies. In some experiments, cells were incubated with the anti-mouse CCR7 antibody (4B12, Biolegend, 120119) for 45 min at room temperature (RT) prior to surface antibody staining.

For intracellular staining, cells were fixed and permeabilised with the Foxp3 Fixation/Permeabilization kit according to manufacturers' instructions (eBioscience, 00-5523-00) for 20 min at RT. Cells were washed and stained for 30 min at 4 °C using the following antibodies: TLR3 (11F8, 141905), GFP (FM264G, 338008) (BioLegend) in permeabilization buffer. In some experiments, permeabilised cells were stained with the anti-cleaved caspase 3 antibody (Cell Signalling Technology, 5A1E, 9664) and detected using a secondary anti-rabbit IgG (Jackson ImmunoResearch, 711-606-152). After staining, cells were washed, resuspended in PBS-FCS and analysed on the LSR Fortessa flow cytometer (BD).

### Cell sorting

Prior to sorting, CCR7$^{gfp/+}$ SI LP cDCs were enriched according to the manufacturer's instructions using MagniSort Streptavidin Negative Selection Beads (Thermo Fisher, MSNB-6002-71). Briefly, after cell isolation, SI LP cells were resuspended at a maximum concentration of $1 \times 10^7$ cells/100 µl in the biotinylated antibody staining mix (BioLegend; B220 (RA3-6B2), CD19 (6D5), CD3 (17A2), CD64 (X54-5/7.1), IgA (RMA-1), Ly6G (1A8)). After incubation for 20 min at 4 °C, cells were washed in 1 mM EDTA in PBS-FCS at 400 x g and 4 °C for 5 min, resuspended at 10$^7$ cells/100 µl. 5 µl of MagniSort streptavidin beads were added per 10$^7$ cells and mixed (for 10 min, RT) on a tube rotator (VWR). 1 ml separation buffer was added and tubes placed on a DynaMag 2 magnet for 5 min. The supernatant, containing the enriched cDC fraction was transferred into new tubes, washed and stained for surface markers with fluorescently labelled antibodies for cell sorting. For single cell sequencing, cDC-enriched SI LP cells from two CCR7$^{gfp/+}$ mice were pooled; 36000 GFP$^-$ and 12000 MHCII$^{hi}$ GFP$^+$ SI LP cDCs were sorted into sterile filtered PBS containing 0.04% bovine serum albumin (BSA) (Miltenyi) using a FACSAria Fusion (BD). The cell concentration was adjusted to 1200 cells/µl in PBS/0.04% BSA for subsequent single cell sequencing library preparation. For single cells sequencing of SI LP cDCs from WT mice, SI LP cells from five female mice were pooled, stained with surface antibodies immediately following isolation and CD11c$^+$ MHCII$^+$ CD64$^-$ B220$^-$ CD19$^-$ CD3$^-$ NK1.1$^-$ cDCs were sorted using a FACSAria IIu (BD).

### In vivo proliferation kinetics

BrdU and EdU proliferation assays were performed according to manufacturers' instructions. In brief, mice were administered 1 mg of BrdU (Sigma Aldrich) or EdU (Baseclick) in 100 µl PBS i.p. 2 h or 24 h later SI and MLNs were excised and cells were isolated according to the protocols above. Afterwards, cells were stained with Zombie NIR (BioLegend, 423105; 1:1000), followed by surface staining and fixation/permeabilization using the Foxp3 Fixation/Permeabilization kit (eBioscience, 00-5523-00).

For BrdU staining, cells were washed in PBS-FCS and resuspended in PBS-FCS containing 300 µg/ml of DNAse I (Roche) for 1 h at 37 °C. Afterwards, cells were washed in permeabilization buffer and stained with anti-BrdU (Bu20a, 339812) and anti-GFP (FM264G, 338008) antibodies (both from Biolegend) in permeabilization buffer for 1 h at RT.

EdU incorporation was detected using the ClickTech EdU Cell Proliferation Kit (BaseClick, BCK-EdU594FC50+IV-S) using a modified

manufacturer's protocol. Briefly, cells were washed once with PBS-FCS after permeabilization. EdU-Click reaction was performed by incubating $1\times10^6$ cells for 30 min at RT in 50 µl of the click reaction buffer containing 0.25 µM of Eterneon-Red 645 Azide. After incubation, cells were washed with PBS-FCS and analysed by flow cytometry.

## Immunohistochemistry

SI segments ( ~ 2 cm) from duodenum, jejunum and ileum were rinsed with PBS-FCS and fixed in Antigenfix (Diapath) for 2 h at 4 °C, followed by a PBS wash, and incubated overnight in PBS with 35% sucrose. The tissues were embedded in optimum cutting temperature (O.C.T) compound (Sakura Finetek, SA62550-01) and stored at −85 °C. For immunostaining, 10 µm sections were cut using LEICA CM3050 S cryostat and collected on pre-treated object slides with VECTABOND reagent (Biozol, VEC-SP-1800). The slides were dried at RT overnight, incubated for 2 h at 56 °C, and rehydrated for 2 min in ddH2O. To reduce autofluorescence, sections were first treated with Image-iT FX (Thermo Fisher, I36933) for 30 min and blocked with PBST (0.05% Tween) with 1% BSA (Carl-Roth, 3737.3) for 2 h at 4 °C to prevent unspecific binding. Sections were stained with anti-mouse TLR3-APC (BioLegend; 11F8, 141905) 1:100 and anti-mouse Ki67-AF488 (eBioscience; SolA15, 53-5698-82) in blocking solution at 4 °C overnight and washed (3x) in PBS for 5 min. Then, sections were stained with DAPI (Carl-Roth; 6843.1; 1 µg/ml) for 3 min and washed (2x) in ddH2O. The slides were embedded in VectaMount AQ Aqueous Mounting Medium (H-5501-60) and imaged using a Zeiss Axio Imager.M2 fluorescent microscope equipped with the Plan-Apochromat 20x/0.80 Ph2 M27 objective.

## Image pre-processing and cDC1 quantification

Background noise in the images was removed using a 50-pixel ball radius filter in Fiji software[52]. TLR3-positive cells were manually annotated as TLR3 staining in close proximity to a DAPI+ nucleus within the LP of intact cross-sectioned villi using QuPath version 0.5[53]. Cells were considered positive for Ki67 if the staining was localised to the DAPI+ nucleus. cDC1 clusters and their sizes were identified by applying Delaunay clustering with a 40 µm threshold.

## Mathematical modelling

Multiple differential equation models were used to study cDC kinetics in the SI. Model 1 is a one-compartment ordinary differential equation (ODE) model which treats the SI as a well-mixed tank. Cells enter from the blood stream at a constant rate, can proliferate locally and exit from the SI. The one-compartment ODE design implies that the probability to proliferate is the same for all cells, independent of the time they have resided in the SI. The same holds for the probability to migrate to the LN. Model 2 is a structured population model which assumes that all cDCs leave the SI after a fixed finite time period. The model is given by a one dimensional transport equation with a proliferation term. Guided by the EdU labelling data, it is assumed that the proliferation rate increases with the time the cDCs reside in the SI. Model 3 assumes that there is a finite upper bound for the time cDCs can spend in the SI. However, cDCs can exit before having reached the maximal residence time. As in Model 2, the proliferation rate increases with the time cDCs spend in the SI. Model 3 is given by a one dimensional transport equation with a proliferation and efflux term. All models were fit the photoconversion data from Fig. 4, assuming multiple plausible choices for the proliferation rate. Details are provided in Supplementary Data 4 and the code used for the models is available at [https://github.com/tstiehl/DC].

## Library preparation and Single-cell RNA sequencing

Chromium Next GEM Chip K (10x Genomics, 1000287) and the Next GEM Single Cell 5′ Kit v2 (10x Genomics, 1000265) were used for GEM generation and library was prepared using the Library Construction Kit (10x Genomics, 1000190) according to the Chromium Next GEM Single Cell 5-v2 User guide (10x Genomics; RevD steps 1, 2 and 5). 50 ng DNA with 14 total cycles were used in Step 5.5. Quality control (QC), quantification and sequencing steps were performed by the Interdisziplinäres Zentrum für Klinische Forschung (IZKF) of the University Hospital RWTH Aachen or the MVLS Shared Research Facilities of the University of Glasgow. For cells from CCR7gfp/+ mice, QC and quantification of cDNA was performed on a high sensitivity D1000 ScreenTape Chip for Tape Station 4200 (Agilent) following the user guide. Library DNA concentration was determined using a Quantus (Promega). Sequencing was performed on a NextSeq 500 sequencing system (Illumina). For cells from WT mice, concentrations were measured using the Qubit HS DNA kit and the libraries were run on the Agilent Bioanalyser (Agilent). Sequencing was performed on a NextSeq 2000 sequencing system (Illumina).

## RNA-sequencing analysis

Raw single-cell sequencing data were processed by CellRanger pipeline with default parameters. The *Mus musculus* reference genome (mm10) was modified to add an additional sequence of the gene encoding *EGFP* [https://www.snapgene.com/plasmids/fluorescent_protein_genes_and_plasmids/EGFP]. The R package Seurat[54] was used to perform downstream analysis: cells having the percentage of mitochondrial >10% and percentage of contaminated ambient RNA >50% were excluded from the analysis as those are indicating dead cells. Doublets were excluded by using DoubletFinder[55]. For dimensionality reduction, a principal component analysis to transform the top 2000 most highly variable genes to 25 principal components was applied. A Uniform Manifold Approximation and Projection (UMAP) transformation followed, which transformed the gene expression profiles of all cells to two dimensions. Clustering was done at cluster resolution of 1. For RNA velocity inference, velocyto[56] and scVelo[57] were used. For pseudotime trajectory analyses monocle 2.12.0[58] was used. Volcano plots were generated using R packages DESeq2[59] and ggplot2 [https://cran.r-project.org/web/packages/ggplot2/citation.html].

## Software and statistical analysis

Flow cytometry data was analysed using FlowJo (BD Life Sciences, version 10.7.2). Statistical analyses were performed as indicated in the figure legends using GraphPad Prism Software (GraphPad Prism 10). P-values ≤ 0.05 were considered significant. Figures and graphical outputs were created using BioRender.com or Adobe Illustrator (Adobe, version 29.1).

## Reporting summary

Further information on research design is available in the Nature Portfolio Reporting Summary linked to this article.

# Data availability

All scRNAseq data generated from experiments in this study have been deposited in the GEO database under the accession number GSE283808. Some of the data used in this study had been previously published and is available under the accession numbers GSE192742, GSE160156 or from [https://www.gutcellatlas.org]. All other data are available in the article and its Supplementary files or from the corresponding author upon request. Source data are provided with this paper.

# Code availability

For mathematical modelling, details are provided in Supplementary Data 4 and the code used for the models is available at [https://github.com/tstiehl/DC].

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

## Acknowledgements

We thank Christina Petrick and Kristina Vukovic Dukic for technical assistance. We are grateful to Alina Viehof and Sareh Tavakol (Institute of Medical Microbiology, University Hospital of RWTH Aachen) for generation of germ-free mice. We thank Dr Ana Izcue and Dr Johanna Kabbert for critical reading of this manuscript and useful discussions. The work was supported by the Flow Cytometry Facility, German Research Foundation (DFG) grant project ID 439895892, and the Genomics Facility, core facilities of the Interdisciplinary Center for Clinical Research (IZKF) Aachen within the Faculty of Medicine at RWTH Aachen University. The work was supported by grants from the DFG: CE 345/1-1 to VC and Project-ID 403224013 – SFB 1382 (B03 to VC).

## Author contributions

Performed experiments and analysed data: F.T.H., As.L., L.K., A.A., S.A.V.J., B.S., An.L. and V.C. Transcriptomic analyses: F.T.H., T.H.N. and L.K. Mathematical modelling: T.S. Methodology: F.T.H., As.L., L.K., A.A., S.A.V.J., An.L., I.P., R.F., O.P. and V.C. Conceived experiments: F.T.H., T.C., S.M., O.P. and V.C. Funding Acquisition: O.P. and V.C. Writing – original draft: F.T.H. and V.C. Writing – review and editing: F.T.H., B.S., S.M., T.C., I.P., R.F., T.S., O.P. and V.C.

## Funding

## Competing interests

The authors declare no competing interests.
