## [Transparent Peer Review file · Nature Communications]

Progressive changes in phenotype, transcriptome and proliferation capacity characterise continued maturation and migration of intestinal cDCs in homeostasis

Corresponding Author: Dr Vuk Cerovic

Version 0:

Reviewer comments:

Reviewer #1

(Remarks to the Author)

In their manuscript, Hager et al. aim to unravel the kinetics and molecular regulation of murine intestinal cDC lifecycle and migration during homeostasis. Using in vivo techniques and molecular characterization they describe molecular changes in the intestinal cDCs from tissue entry, differentiation, and CCR7 upregulation, to migration to the intestine draining mesenteric lymph nodes. The authors demonstrate that the life cycle of intestinal cDC is highly dynamic and has an estimated daily turnover between 14.4-26.3%.

The results presented are very interesting and specifically address healthy steady-state intestinal cDC biology which is key to better understanding how cDC contribute to continuous immune decision-making to maintain tolerance to harmless antigens while retaining the capacity to generate host defense to pathogenic antigens. The experiments are technically challenging and well-performed. The resulting data are presented clearly, and the manuscript is well-written. In general, the conclusions are supported by the provided data. However, there are several concerns.

1. The scRNAseq data presented in Figure 1B were generated by pooling sorted SI LP cDC from n=2 CCR7gfp/+ mice. As this analysis forms a key element in the paper and such a dataset is likely to be reused after publication, it is important to know how reproducible this data is between mice, to assure that n=2 is sufficient.
2. In figure 2I, the representative immunofluorescence shows TLR3 staining (red) in an area where there is no tissue (dapi). The TLR3 staining should be checked. Please provide a zoom-in in the top corner to get a better view of the clusters.
3. Figure 3H and I are not convincing. These figures have no statistics (also missing in panel D), are based on n=3 (presumably with SEM? Please indicate). On top of this, for several molecules, the Y-axis do not start at MFI=0, questioning whether there is a real relevant difference. No representative dotplots are shown. These data need to be strengthened.
4. The discussion is very long and is sometimes a bit suggestive. Shortening to emphasize key points is advised.

Examples of suggestive phrases/overstatements:

- Line 378 “pre-migratory cDC2s expressed higher levels of Il1b reflecting their proposed role in intestinal Th17 responses”. This is a stretch considering that Il1b mRNA does not always agree with protein production and IL-1 activity requires inflammasome activation.
- Line 408 While this hints that ...cessation of proliferation may act as one of the signals ...homeostatic activation and cDC migration. Very suggestive.

(Remarks on code availability)

Reviewer #2

(Remarks to the Author)

Classical DC are critical players for gut immune homeostasis and in depth understanding and turn-over rates of distinct DC subpopulation could be important for the development of vaccination strategies. Here, Hager et al. use state-of-the-art mouse

models combined with single RNAseq profiling to provide a comprehensive analysis of the intestinal classical DC compartment. Specifically, they take advantage of a newly established mouse strain, i.e. CCR7-GFP mice, that allows them to dissect pre-migratory DC and cells which are about to leave the lamina propria to relocate to mesenteric LNs. The authors make a number of interesting observations, with the most exciting and novel being (1) the robust DC turnover kinetics in lamina propria and LNs and (2) the discovery of profound proliferative expansion of DC in the tissue context that precedes a maturation continuum and emigration (also recently revealed by fate mapping by another team). Using Vav-H2B-Dendra2, the authors elegantly determine the site of proliferation, as well as the residence time of distinct DC subsets. Taken together, this is a very interesting and informative study, the conclusions of which are substantiated by solid experimental evidence. Provided the authors fix some of the shortcomings outlined below, I support publication in Nat comm.

Major points

1) Fig. 1 please mention for better understanding in the corresponding text that you sorted MHC-II positive CD64-negative cells for the sc analysis.

The authors introduce CCR7-gfp reporter mice but the information provided on this strain is too cryptic to fully appreciate the model. A schematic of the construct and some basic characterization of the expression profile in lymphoid and non-lymphoid tissue, as well as blood, should be added. I understand the CCR7-gfp insertion generates a null allele. This should be more clearly stated. It also introduces a certain caveat as the scRNAseq data presented in Fig. 1 are inherently from heterozygote mutant CCR7 mice, not C57BL/6 animals. The data probably stand, but this should be acknowledged.

2) The authors talk about homeostatic turnover, but arguably provide data on two tissues that are rather unique given the exposure to microbial stimuli. This should be better acknowledged. The authors touch on this aspect in the discussion and refer to prior studies. However, as they have the new model, it could potentially be interesting to gauge the impact of endotoxin exposure by analyzing germ-free animals or exposure to antibiotics and see if the turnover rate changes. Did the authors analyze the LNs that drain distinct gut segments separately (see 10.1038/s41586-019-1125-3) ? The inclusion of such a data set, in particular, if differences are observed, could be a valuable addition, also for a better understanding of microbiome impact.

3) In Figures 3H and 3I, the authors attempt to distinguish cDC2a and cDC2b using flow cytometry analysis. However, it is unclear whether the expression level of CD103 is sufficient to differentiate these subsets. Consider using the T-bet to classify discriminate cDC2As and cDC2B.

4) Notably, the violin plot in Figure 3G indicates that the expression level of Itgae (encoding CD103) does not appear to show a significant difference across clusters. Additional clarification or supporting data is needed to substantiate this distinction.

5) In Figure 2I, the authors suggest that the cDC1 population proliferates in tissue based on histological data. However, the proximity shown in the figure alone does not provide direct evidence of proliferation. To strengthen this claim, the authors should perform Ki67 staining and demonstrate a correlation between Ki67 expression and the observed proximity.

Minor points

1) I suggest to change in the abstract the reference to percentages with 'up to a quarter'. Likewise for clarity consider to rephrase to: 'an almost complete daily replacement of the migratory cDC compartment in the mesenteric LN'.

2) Please provide representative FACS plots for Figures 2K, 3D, 3H, and 3I to enhance clarity and support the presented data.

(Remarks on code availability)

Reviewer #3

(Remarks to the Author)

Although several cDC subsets, such as cDC1, CD103+ cDC2, and CD103- cDC2, with distinct features in the human and mouse small intestine, have been described in many studies, the kinetics of their turnover and mechanisms regulating their migration and life span remain poorly understood. The manuscript by Hager et al. analyzes cDC subsets in the small intestinal lamina propria (SI LP) and MLN using scRNA-seq combined with photoconversion. The authors propose continuous alterations in transcriptional profiles, surface expression levels of MHC class II, and proliferative capacities in cDC1s and cDC2s in the SI LP under homeostatic conditions. The authors highlight the kinetics of turnover of cDC1s and cDC2s in the SI LP and MLN, and they also show that an S1P modulator FTY720 decreases the migration of CD103+ cDC2s to the MLP, but not that of cDC1s and CD103- cDC2s. The quantitative method to assess cDC turnover and the state-of-the-art techniques used in this manuscript are highly appreciated. However, the quality of the scRNA-seq data sets is insufficient to demonstrate reproducibility, which limits the impact of the study. Moreover, most of the findings are in line with known cDC intestinal biology, but the study remains descriptive and provides no functional tests to support the authors' interpretation, and it lacks some novelty. Overall, the main idea of the study and some of the findings presented in this study are interesting, little direct evidence is provided for these mechanistic connections.

Main comments

(1) FACS-enriched cDCs are divided into 12 populations, including cDC1 populations (0, 7, and 8), cDC2 populations (1, 2,

3, 4, 5, 6, and 10), and pre-migratory cDCs (9), based on transcriptome features. The authors state that sub-population 7 and sub-population 2 are immature cycling cDC1s and cDC2s, respectively. However, the hypothesis is not supported by any evidence other than results from scRNA-seq data sets. The authors should carry out adoptive transfer experiments to determine whether these populations have proliferative capacity in the small intestine and their progeny matures into pre-migratory cDC1s and cDC2.

(2) In Figures 1 and 3, the authors describe changes in gene expression of co-stimulatory molecules, cell cycle-related molecules, and pro/anti-apoptotic molecules as well as cDC markers in sub-populations of cDCs. It should be examined whether protein expression levels of these molecules are also altered in these cells because the surface expression level of MHCII (Figure 1A) is inconsistent with the mRNA expression levels of H2-Aa and H2-Ab1 in both CCR7- cDCs and pre-migratory cDCs (Figure 1D).

(3) In Figure 2I, the immunohistochemistry performed on the small intestine shows that TLR3+ cells (cDC1s) form clusters of up to 10 cells, in accordance with findings showing that part of cDC1s incorporate EdU in the SI LP. The authors should demonstrate cluster formation of cDC2s in the SI LP via spatial transcriptomics (or immunohistochemistry).

(4) In Figures 2L and 2M, the authors show the M3 and M4 subset in cDC1s and cDC2s contain proliferative cells in the SI LP, while CCR7+ pre-migratory cDCs lose their proliferative capacity. Which sub-clusters of cDCs (shown in Figure 1B) are contained in the M3 and M4 subset should be explored by performing scRNA-seq or bulk RNA-seq. Also, are the M4 cells with high proliferative capacity composed of sub-population 7 cDC1s and sub-population 2 cDC2s (shown in Figure 1B)?

(5) Which signals induce and suppress the local proliferation of each cDC population in the SI LP? The proliferative activity of CCR7- cDCs and pre-migratory cDCs in the SI LP of CCR7(gfp/gfp) mice should be analyzed to determine whether CCR7 signaling is involved in regulation of cell proliferation in cDC populations. Additionally, the authors should investigate proliferative cells in each cDC population in the SI LP of Csf2r-/- and Flt3-/- mice, as Flt3L and CSF2 have been identified to induce cDC development and maintain cDC homeostasis (ref39, 40).

(6) Several studies have demonstrated that cDCs drive differentiation of effector T cells and regulatory T cells, as the authors state in Discussion of the manuscript. Could the authors analyze the ability of the M1, M2, M3, and M4 subsets and pre-migratory cDCs within small intestinal cDC1s and cDC2s to initiate development of Th1, Th17, and Foxp3+ regulatory T cells to determine whether only pre-migratory cDCs can induce T cell differentiation?

(7) In Figure 5C, the authors show that the relative percentages of cDC1s and cDC2s present in the MLN during photoconversion process (Dred+ cells) are rapidly reduced and describe "the short lifespan of the migratory cDCs in the MLN is consistent with the upregulation of apoptotic genes in pre-migratory cDCs present in the SI LP" in the manuscript. Whether cDCs undergo apoptotic cell death in the MLN should be examined by conducting TUNEL staining to support their claim.

(8) The authors should analyze what environmental factors affect the turnover rate/life span of each cDC subset in the MLN and SI LP under homeostatic conditions at least using Csf2r-/- and Flt3-/- Vav-H2B-Dendra2 mice, as Flt3L and CSF2 have been reported to contribute to development of cDC and maintenance of their homeostasis (ref39, 40).

(9) In Figure 5E, FTY treatment increases the percentage of Dred+ cells in CD103+ cDC2s, but not cDC1s and CD103- cDC2s, in the photoconverted MLN. The authors should analyze the total number of CD103+ cDC2s, CD103- cDC2s, and cDC1s in the MLN and SI LP of mice treated with or without FTY to validate that FTY specifically inhibits the migration of CD103+ cDC2s from the SI LP to the MLN. In addition, the authors should compare the transcript levels of S1P receptors among cDC1, CD103+ cDC2s, and CD103- cDC2s in the SI LP to define why FTY has a cell type-specific effect.

(10) Regarding the above, the authors should analyze surface expression of CCR7 on MHCII(high) cDC1, CD103+ cDC2s, and CD103- cDC2s in the SI LP of FTY-treated mice because a previous study (PMID: 26416269) reported that FTY decreases CCR7 expression on CD11c+ DC and inhibits DC migration into the dLN.

Minor comments

- (1) Please include a list of DEG in sub-clusters of cDC1s, CD103+ cDC2, and CD103-cDC2s isolated from the SI LP.
- (2) Please show the gating strategy for B cells (Supplementary Figure 5C).
- (3) Page 7 line 150: please change (Supplementary Fig. 1C,) to (Supplementary Fig. 1C).

(Remarks on code availability)

Version 1:

Reviewer comments:

Reviewer #1

(Remarks to the Author)

The authors have revised the manuscript according to the reviewers' suggestions. This has improved the manuscript a lot. I

have no further comments.

(Remarks on code availability)
Have not reviewed the code.

Reviewer #2

(Remarks to the Author)
The authors have answered all my concerns. This is a very informative and comprehensive study.

(Remarks on code availability)

Reviewer #3

(Remarks to the Author)
The authors have addressed and thoroughly discussed my main concerns in the revised manuscript.

(Remarks on code availability)

We would like to thank all three reviewers for their helpful and constructive comments. We have now included substantial new data and made changes to the text of the manuscript. A full list of changes is listed in the point-by-point response and highlighted in the revised manuscript, but a brief summary of the major changes is listed here.

- Included an independently performed scRNAseq dataset from WT SI LP DCs, as a validation cohort for our analysis of CCR7^{gfp/+} mice (Figure 1F and new Supplementary Fig. 2)
- Added data demonstrating that functional CCR7 expression is not required for the regulation of SI LP cDC proliferation (new Figure 2N)
- Added staining for cleaved caspase 3, demonstrating increased apoptosis among migratory MLN DCs (Fig. 5E)
- Added data from SI LP and MLN cDC of germ-free mice, demonstrating that major parameters of cDC turnover are unaffected by lack of the gut microbiota (Supplementary Fig. 6C and D)
- Increased the number of replicates and included FACS histograms of protein expression along a cDC maturation continuum (Figure 3)

Reviewer's Comments:

Reviewer #1 (Remarks to the Author)

In their manuscript, Hager et al. aim to unravel the kinetics and molecular regulation of murine intestinal cDC lifecycle and migration during homeostasis. Using in vivo techniques and molecular characterization they describe molecular changes in the intestinal cDCs from tissue entry, differentiation, and CCR7 upregulation, to migration to the intestine draining mesenteric lymph nodes. The authors demonstrate that the life cycle of intestinal cDC is highly dynamic and has an estimated daily turnover between 14.4-26.3%.

The results presented are very interesting and specifically address healthy steady-state intestinal cDC biology which is key to better understanding how cDC contribute to continuous immune decision-making to maintain tolerance to harmless antigens while retaining the capacity to generate host defense to pathogenic antigens. The experiments are technically challenging and well-performed. The resulting data are presented clearly, and the manuscript is well-written. In general, the conclusions are supported by the provided data. However, there are several concerns.

We thank the reviewer for their positive comments and helpful suggestions. We have now included substantial additional data and made edits to the manuscript which we hope address the reviewer's concerns, resulting in an improved manuscript.

1. The scRNAseq data presented in Figure 1B were generated by pooling sorted SI LP cDC from n=2 CCR7^{gfp/+} mice. As this analysis forms a key element in the paper and such a dataset is likely to be reused after publication, it is important to know how reproducible this data is between mice, to assure that n=2 is sufficient.

We agree that the scRNAseq data of intestinal cDCs likely represents a key dataset of our manuscript and is likely to benefit other researchers. To ensure reproducibility, we have now included an additional scRNAseq dataset from SI LP cDCs of 5 pooled WT mice as a "validation cohort". The data, in the new Supplementary Figure 2 show similar clustering, with major clusters of cDC1s, cDC2s and

pre-migratory CCR7⁺ cDCs. Moreover, the CCR7⁺ WT and CCR7^{gfp/+} SILP DCs show a substantial overlap of DEGs and high similarity score (Figure 1F).

We think that the additional data addresses the concerns about the reproducibility of the presented scRNAseq dataset and further strengthens the validity of our conclusions.

Reviewer Figure: UMAP analysis of scRNAseq from CCR7^{gfp/+} mice (left) and WT mice (right)

2. In figure 2I, the representative immunofluorescence shows TLR3 staining (red) in an area where there is no tissue (dapi). The TLR3 staining should be checked. Please provide a zoom-in in the top corner to get a better view of the clusters.

A zoomed-in section of the staining has now been included in Figure 2I.

We acknowledge that there are areas of spurious staining visible in the lumen of the gut. In our experience, this is a relatively frequent occurrence in immunohistochemistry of gut sections, caused by the “stickiness” of gut content. To ensure that only bona fide cells were counted in the analysis, we assigned TLR3+ cell identity only to staining localised in the LP and only when in close proximity to a DAPI+ nucleus. This point has been added to the Methods section.

3. Figure 3H and I are not convincing. These figures have no statistics (also missing in panel D), are based on n=3 (presumably with SEM? Please indicate). On top of this, for several molecules, the Y-axis do not start at MFI=0, questioning whether there is a real relevant difference. No representative dotplots are shown. These data need to be strengthened.

We have now performed additional measurements, increasing the number of replicates to n=7 performed on two separate occasions (represented as normalised MFI to account for inter-experimental variation). We also added representative histograms of the FACS staining and performed a statistical analysis in Figure 3 panels D, H and I, highlighting significant changes in the indicated markers across the DC maturation continuum. The data in panels 3D, 3H and 3I are presented as mean ± SD, which is now also specified in the figure legend.

4. The discussion is very long and is sometimes a bit suggestive. Shortening to emphasize key points is advised.

Examples of suggestive phrases/overstatements:

- Line 378 “pre-migratory cDC2s expressed higher levels of Il1b reflecting their proposed role in intestinal Th17 responses”. This is a stretch considering that Il1b mRNA does not always agree with protein production and IL-1 activity requires inflammasome activation.
- Line 408 While this hints thatcessation of proliferation may act as one of the signals ...homeostatic activation and cDC migration. Very suggestive.

The discussion has been amended in line with the reviewer’s suggestions.

(Remarks on code availability)

Reviewer #2 (Remarks to the Author)

Classical DC are critical players for gut immune homeostasis and in depth understanding and turn-over rates of distinct DC subpopulation could be important for the development of vaccination strategies. Here, Hager et al. use state-of-the-art mouse models combined with single RNAseq profiling to provide a comprehensive analysis of the intestinal classical DC compartment. Specifically, they take advantage of a newly established mouse strain, i.e. CCR7-GFP mice, that allows them to dissect pre-migratory DC and cells which are about to leave the lamina propria to relocate to mesenteric LNs.

The authors make a number of interesting observations, with the most exciting and novel being (1) the robust DC turnover kinetics in lamina propria and LNs and (2) the discovery of profound proliferative expansion of DC in the tissue context that precedes a maturation continuum and emigration (also recently revealed by fate mapping by another team). Using Vav-H2B-Dendra2, the authors elegantly determine the site of proliferation, as well as the residence time of distinct DC subsets.

Taken together, this is a very interesting and informative study, the conclusions of which are substantiated by solid experimental evidence. Provided the authors fix some of the shortcomings outlined below, I support publication in Nat comm.

We thank the reviewer for their positive assessment of the manuscript and the constructive comments and suggestions. We have now included substantial additional data and made several modifications to the text to address the reviewer’s concerns.

Major points

1) Fig. 1 please mention for better understanding in the corresponding text that you sorted MHC-II positive CD64-negative cells for the sc analysis.

This point has now been further emphasised in the results section in addition to pointing the reader to the gating strategy in Supplementary Figure 1C.

The authors introduce CCR7-gfp reporter mice but the information provided on this strain is too cryptic to fully appreciate the model. A schematic of the construct and some basic characterization of the expression profile in lymphoid and non-lymphoid tissue, as well as blood, should be added. I understand the CCR7-gfp insertion generates a null allele. This should be more clearly stated. It also introduces a certain caveat as the scRNAseq data presented in Fig. 1 are inherently from heterozygote mutant CCR7 mice, not C57BL/6 animals. The data probably stand, but this should be acknowledged.

We apologise from the insufficient description of the CCR7-GFP mice. A map of the genetic construct, as well as the characterisation of the CCR7 protein and GFP expression on blood T cells have been added to the new panels in Supplementary figures 1A and 1B. We also amended the text on page 5 of the Results and on page 21 in the Methods section to give more detail on the CCR7-GFP mice.

The reviewer raises an important point, that the lack of one functional copy of the CCR7 gene in CCR7^{gfp/+} mice may potentially impact the gene expression detected in our scRNAseq experiment. To address this possibility, we have now included an additional scRNAseq dataset from SILP cDCs of 5 pooled WT mice as a “validation cohort”. The data, included in Supplementary Figure 2A-C, and Figure 1F show similar clustering and key differentially regulated genes as our initial analysis of CCR7^{gfp/+} mice. We believe these additional data increase the validity of the presented scRNAseq dataset.

Reviewer Figure: UMAP analysis of scRNAseq from CCR7^{gfp/+} mice (left) and WT mice (right)

2) The authors talk about homeostatic turnover, but arguably provide data on two tissues that are rather unique given the exposure to microbial stimuli. This should be better acknowledged. The authors touch on this aspect in the discussion and refer to prior studies. However, as they have the new model, it could potentially be interesting to gauge the impact of endotoxin exposure by analyzing germ-free animals or exposure to antibiotics and see if the turnover rate changes. Did the authors analyze the LNs that drain distinct gut segments separately (see 10.1038/s41586-019-1125-3) ? The inclusion of such a data set, in particular, if differences are observed, could be a valuable addition, also for a better understanding of microbiome impact.

We agree with the reviewer that the differences between colon and SI may provide crucial insights in the processes controlling DC migration. While we did not analyse the individual SI-draining MLNs, we did analyse the SI and colon- draining nodes separately (Figure 5C, Supplementary Fig. 6B). We observed essentially no difference in the turnover rates of either the migratory or resident MLN compartment between the MLNs draining the SI or the colon.

Previously published data (PMID: 26259586, PMID: 18026177) suggest that overall proportion of MLN DCs is not altered regardless of microbial colonisation status. In line with this, we observed no difference in the proportion of pre-migratory CCR7⁺ SI LP cDCs between germ-free (GF) and specific

pathogen free (SPF) mice. Moreover, additional analysis of unpublished data collected as part of our previous study (PMID: 36208631) showed no difference in the ratio of migratory to resident cDCs in the MLN of GF and SPF mice. The data have now been included in the manuscript (Supplementary Fig. 6C, D).

Reviewer Figure: the proportion of CCR7+ SI LP cDCs in GF and SPF mice (left) and the ratio of migratory to resident cDCs in the MLN of GF and SPF mice.

Collectively, these data lead us to conclude that it is unlikely that microbial burden plays a crucial role in homeostatic DC migration from either the SI or the colon. We have extended the discussion of the possible role of microbial stimuli in intestinal cDC migration and homeostasis.

3) In Figures 3H and 3I, the authors attempt to distinguish cDC2a and cDC2b using flow cytometry analysis. However, it is unclear whether the expression level of CD103 is sufficient to differentiate these subsets. Consider using the T-bet to classify discriminate cDC2As and cDC2B.

The author raises an important point as intestinal surface markers do not delineate cDC2a and cDC2b lineages perfectly. Unfortunately, in our hands, we could not detect T-bet by flow cytometry on intestinal cDCs. Moreover, expression of the *Tbx21* gene was confined to a small number of cDC2s which did not coincide with any of the predefined clusters.

Reviewer Figure: *Tbx21* expression among sorted SI LP cDCs by scRNAseq

We have changed the labels in figure 3H and 3I to refer to the protein expression on CD103- and CD103+ subsets of cDC2s specifically and do not comment on their lineage. We further emphasise in the results that the CD103 staining does not perfectly correlate with cDC2a and cDC2b lineages and highlight the associated caveats.

4) Notably, the violin plot in Figure 3G indicates that the expression level of *Itgae* (encoding CD103) does not appear to show a significant difference across clusters. Additional clarification or supporting data is needed to substantiate this distinction.

We also noted that the expression levels of *Itage* mRNA do not usefully differentiate CD103+ and CD103- cDC2s (even though both populations were present in the sorted sample). However, low expression of *Itgb7*, which encodes the integrin beta7 subunit of CD103 was observed in cluster 4 among total cDCs in Figure 1. After re-clustering of cDC2s in figure 3, we noted that expression of *Itgb7* was lower or absent in the new clusters 5 and 6, which also expressed genes characteristic of cDC2bs. We have now added a violin plot of *Itgb7* expression to figure 3G. We also added a remark in the results section to clarify that we observe a correlation between CD103- DCs and the DC2b transcriptomic signature but that there are caveats inherent in our identification of the cDC2 subtypes in the intestine.

5) In Figure 2I, the authors suggest that the cDC1 population proliferates in tissue based on histological data. However, the proximity shown in the figure alone does not provide direct evidence of proliferation. To strengthen this claim, the authors should perform Ki67 staining and demonstrate a correlation between Ki67 expression and the observed proximity.

We thank the reviewer for this suggestion. Microscopy pictures of Ki67 staining in cDC1 LP clusters have now been included in Supplementary Figure 3B. Although Ki67 staining was largely prevalent among cDC1s in the LP, Ki67- cells were more likely to be observed in small clusters or solitary TLR3+ cells, whereas all observed clusters of five or more cells always included Ki67+ cells and over 70% of cells in large (5+) clusters were Ki67+. This information has now been included in the results section.

Minor points

1) I suggest to change in the abstract the reference to percentages with 'up to a quarter'. Likewise for clarity consider to rephrase to: 'an almost complete daily replacement of the migratory cDC compartment in the mesenteric LN'.

The abstract has been changed in line with this suggestion.

2) Please provide representative FACS plots for Figures 2K, 3D, 3H, and 3I to enhance clarity and support the presented data.

Representative overlay histograms have been added to panels 2K, 3D, 3H and 3I. The overall number of replicates in 3D, 3H and 3I has been increased to n=7.

(Remarks on code availability)

Reviewer #3 (Remarks to the Author)

Although several cDC subsets, such as cDC1, CD103+ cDC2, and CD103- cDC2, with distinct features in the human and mouse small intestine, have been described in many studies, the kinetics of their turnover and mechanisms regulating their migration and life span remain poorly understood. The manuscript by Hager et al. analyzes cDC subsets in the small intestinal lamina propria (SI LP) and MLN using scRNA-seq combined with photoconversion. The authors propose continuous alterations in transcriptional profiles, surface expression levels of MHC class II, and proliferative capacities in cDC1s and cDC2s in the SI LP under homeostatic conditions. The authors highlight the kinetics of turnover of cDC1s and cDC2s in the SI LP and MLN, and they also show that an S1P modulator FTY720 decreases the migration of CD103+ cDC2s to the MLP, but not that of cDC1s and CD103- cDC2s. The quantitative method to assess cDC turnover and the state-of-the-art techniques used in this manuscript are highly appreciated. However, the quality of the scRNA-seq data sets is insufficient to demonstrate reproducibility, which limits the impact of the study. Moreover, most of the findings are in line with known cDC intestinal biology, but the study remains descriptive and provides no functional tests to support the authors' interpretation, and it lacks some novelty. Overall, the main idea of the study and some of the findings presented in this study are interesting, little direct evidence is provided for these mechanistic connections.

We thank the reviewer for highlighting the state-of-the-art techniques as well as that the study idea and the data within are interesting. We respectfully disagree however, that the data lacks novelty. To our knowledge, this is the first comprehensive analysis of cDC migration kinetics in the gut as well as the first detailed single-cells transcriptomic analysis. While some of the data are descriptive, we hope that the amended manuscript, including substantial new data, addresses most of the reviewer's concerns.

Main comments

(1) FACS-enriched cDCs are divided into 12 populations, including cDC1 populations (0, 7, and 8), cDC2 populations (1, 2, 3, 4, 5, 6, and 10), and pre-migratory cDCs (9), based on transcriptome features. The authors state that sub-population 7 and sub-population 2 are immature cycling cDC1s and cDC2s, respectively. However, the hypothesis is not supported by any evidence other than results from scRNA-seq data sets. The authors should carry out adoptive transfer experiments to determine whether these populations have proliferative capacity in the small intestine and their progeny matures into pre-migratory cDC1s and cDC2.

We did not mean to imply that clusters 2 and 7 are immature cycling DCs. This was our initial hypothesis, suggested by the scRNAseq dataset, as the reviewer notes. However, when we directly tested this hypothesis using photoconversion, we actually found the opposite – that the newly incoming cDCs (Dendra red negative) show a lower proliferation capacity compared to the DCs present in the tissues (Figure 2G). We therefore interpret this data to mean that clusters 7 and 2 actually represent cycling cDCs at varied stages of maturation/development but cluster together due to the proliferation-associated shift in transcriptome. We apologise we did not make this clear in the text itself and have now amended the relevant paragraphs to make this point more clearly.

(2) In Figures 1 and 3, the authors describe changes in gene expression of co-stimulatory molecules, cell cycle-related molecules, and pro/anti-apoptotic molecules as well as cDC markers in sub-populations of cDCs. It should be examined whether protein expression levels of these molecules are also altered in these cells because the surface expression level of MHCII (Figure 1A) is inconsistent

with the mRNA expression levels of H2-Aa and H2-Ab1 in both CCR7⁻ cDCs and pre-migratory cDCs (Figure 1D).

We fully agree with the reviewer that mRNA expression does not necessarily reflect protein expression, as evidenced by MHCII on maturing DCs, which may largely be accounted for by the previously reported internal stores of MHCII on immature/maturing cDCs (PMID: 10727455, PMID: 9285592). We have now added a sentence in the results to highlight this discrepancy and include the relevant citations.

We sought to verify the protein expression of several key gene products, which is presented in Figure 3 on cDC1 and cDC2 separately. To strengthen this part of the manuscript, we performed additional measurements to increase the number of biological replicates from 3 to 7 and performed statistical analysis to highlight significant changes in protein expression along the cDC maturation continuum.

(3) In Figure 2I, the immunohistochemistry performed on the small intestine shows that TLR3⁺ cells (cDC1s) form clusters of up to 10 cells, in accordance with findings showing that part of cDC1s incorporate EdU in the SI LP. The authors should demonstrate cluster formation of cDC2s in the SI LP via spatial transcriptomics (or immunohistochemistry).

We agree that this it would be beneficial to confirm that the clonal clustering we observed cDC1 can also be seen for cDC2s. Unfortunately, we were not able to devise a strategy to reliably identify cDC2s by microscopy (in particular to distinguish them from the very numerous macrophage population). We have added a sentence in the results to emphasise that this finding only encompasses cDC1s.

(4) In Figures 2L and 2M, the authors show the M3 and M4 subset in cDC1s and cDC2s contain proliferative cells in the SI LP, while CCR7⁺ pre-migratory cDCs lose their proliferative capacity. Which sub-clusters of cDCs (shown in Figure 1B) are contained in the M3 and M4 subset should be explored by performing scRNA-seq or bulk RNA-seq. Also, are the M4 cells with high proliferative capacity composed of sub-population 7 cDC1s and sub-population 2 cDC2s (shown in Figure 1B)?

It was not our intention to suggest that bins M1-M4 represent discrete subsets of maturing cDCs in the LP. Rather, we sought to utilise the surface MHCII expression as a surrogate marker of the DC stage of development. While we cannot draw a 1:1 relationship between the maturation bins and the transcriptional changes, we demonstrate that the expression of several key genes and their products follow the same pattern, demonstrating a gradual change in cDC phenotype in tissues and a progressive increase in proliferative capacity, followed by a cessation of proliferation amongst CCR7⁺ pre-migratory DCs. We have now amended the text to make this limitation of the study more clear.

(5) Which signals induce and suppress the local proliferation of each cDC population in the SI LP? The proliferative activity of CCR7⁻ cDCs and pre-migratory cDCs in the SI LP of CCR7(gfp/gfp) mice should be analyzed to determine whether CCR7 signalling is involved in regulation of cell proliferation in cDC populations. Additionally, the authors should investigate proliferative cells in each cDC population in the SI LP of Csf2r^{-/-} and Flt3^{-/-} mice, as Flt3L and CSF2 have been identified to induce cDC development and maintain cDC homeostasis (ref39, 40).

We thank the reviewer for raising this interesting and important question. As suggested, in order to directly assess the proliferation of DCs deficient for CCR7, we analysed proliferation within the M1-M4 and CCR7⁺ maturation bins in SILP DCs of CCR7^{gfp/gfp} mice. The data, included in a new panel in Figure 2N, show a similar trend to DCs from the SI LP of WT and CCR7gfp/+ mice (Figure 2M),

suggesting that CCR7 signalling itself does not play a direct role in the cessation of proliferation seen in pre-migratory DCs.

Reviewer Figure: The frequency of EdU+ cDCs, cDC1s or cDC2s along the maturation bins in WT and CCR7^{gfp/+} mice (top) and CCR7^{gfp/gfp} mice (bottom)

We agree fully with the reviewer that the exact mechanism by which local proliferation of cDCs in intestinal tissue is regulated will be a key question moving forward and one we are keen to address. However, these experiments will require the generation of complex transgenic systems to fully elucidate. Therefore, we consider them to fall outside the scope of the work presented here.

(6) Several studies have demonstrated that cDCs drive differentiation of effector T cells and regulatory T cells, as the authors state in Discussion of the manuscript. Could the authors analyze the ability of the M1, M2, M3, and M4 subsets and pre-migratory cDCs within small intestinal cDC1s and cDC2s to initiate development of Th1, Th17, and Foxp3+ regulatory T cells to determine whether only pre-migratory cDCs can induce T cell differentiation?

We agree with the reviewer that this is a fascinating question and may indeed shed light on the acquisition of antigen presentation capacity during cDC development in the SILP. However, we consider them to be a likely direction of our future research, but to fall outside the scope of the present study.

(7) In Figure 5C, the authors show that the relative percentages of cDC1s and cDC2s present in the MLN during photoconversion process (Dred+ cells) are rapidly reduced and describe “the short lifespan of the migratory cDCs in the MLN is consistent with the upregulation of apoptotic genes in pre-migratory cDCs present in the SI LP” in the manuscript. Whether cDCs undergo apoptotic cell death in the MLN should be examined by conducting TUNEL staining to support their claim.

We thank the reviewer for this excellent suggestion. In addition to the scRNAseq data which show an increase in proapoptotic transcripts in CCR7+ SILP DCs (Figure 1D) we have now included data showing staining for Cleaved caspase 3 in MLN DCs, demonstrating the significantly higher proapoptotic signals in the migratory MLN DCs compared to the resident cDC compartment (Fig. 5E).

Reviewer Figure: Proportion of cDCs staining for cleaved caspase 3 in the migratory and resident MLN cDC compartment.

(8) The authors should analyze what environmental factors affect the turnover rate/life span of each cDC subset in the MLN and SI LP under homeostatic conditions at least using *Csf2r*^{-/-} and *Flt3*^{-/-} *Vav-H2B-Dendra2* mice, as *Flt3L* and *CSF2* have been reported to contribute to development of cDC and maintenance of their homeostasis (ref39, 40).

As we state above, we fully agree with the reviewer that the factors regulating local survival, proliferation and homeostasis of cDCs in tissues is a fascinating question and will require detailed follow-up studies using some of the methodology we present here. However, we consider them to fall outside the scope of the presented work but will likely form the basis of our future work.

(9) In Figure 5E, FTY treatment increases the percentage of Dred⁺ cells in CD103⁺ cDC2s, but not cDC1s and CD103⁻ cDC2s, in the photoconverted MLN. The authors should analyze the total number of CD103⁺ cDC2s, CD103⁻ cDC2s, and cDC1s in the MLN and SI LP of mice treated with or without FTY to validate that FTY specifically inhibits the migration of CD103⁺ cDC2s from the SI LP to the MLN. In addition, the authors should compare the transcript levels of S1P receptors among cDC1, CD103⁺ cDC2s, and CD103⁻ cDC2s in the SI LP to define why FTY has a cell type-specific effect.

We agree that the mechanism behind the selective requirement for S1P signalling for the migration of intestinal CD103⁺ cDC2s but not cDC1s is an intriguing question. We could not detect the expression of any S1PR mRNAs in our scRNAseq dataset. However, genes encoding S1PR1, S1PR3, S1PR4 and S1PR5 were all upregulated in migrating lymph DCs compared to SI LP DCs (data is now highlighted in Supplementary Figure 2D). However, no differences in S1PR expression between cDC subsets could be observed. It is therefore unlikely that this accounts for the subsets-specific inhibition of migration in response to FTY720. We have added a sentence in the Results section to emphasise this point.

(10) Regarding the above, the authors should analyze surface expression of CCR7 on MHCII(high) cDC1, CD103⁺ cDC2s, and CD103⁻ cDC2s in the SI LP of FTY-treated mice because a previous study (PMID: 26416269) reported that FTY decreases CCR7 expression on CD11c⁺ DC and inhibits DC migration into the dLN.

We thank the reviewer for bringing this paper to our attention. We have now included a citation to it in the discussion as a possible mechanism of FTY action on migrating DCs.

Minor comments

(1) Please include a list of DEG in sub-clusters of cDC1s, CD103+ cDC2, and CD103-cDC2s isolated from the SI LP.

The full list of DEGs for all clusters has been incorporated into the new version of Supplementary Table 1.

(2) Please show the gating strategy for B cells (Supplementary Figure 5C).

A FACS plot showing the gating strategy for B cells has been incorporated in the existing DC gating strategy in Figure 5A.

(3) Page 7 line 150: please change (Supplementary Fig. 1C,) to (Supplementary Fig. 1C).

This has now been corrected.

(Remarks on code availability)